# Transient disome complex formation in native polysomes during ongoing protein synthesis captured by cryo-EM

Timo Flügel ®[1,5], Magdalena Schacherl[1,5], Anett Unbehaun[1], Birgit Schroeer[1], Marylena Dabrowski[1], Jörg Bürger[1,2], Thorsten Mielke ®[2], Thiemo Sprink[3,4], Christoph A. Diebolder ®[3,4], Yollete V. Guillén Schlippe ®[1] ✉ & Christian M. T. Spahn ®[1] ✉

Structural studies of translating ribosomes traditionally rely on in vitro assembly and stalling of ribosomes in defined states. To comprehensively visualize bacterial translation, we reactivated ex vivo-derived *E. coli* polysomes in the PURE in vitro translation system and analyzed the actively elongating polysomes by cryo-EM. We find that 31% of 70S ribosomes assemble into disome complexes that represent eight distinct functional states including decoding and termination intermediates, and a pre-nucleophilic attack state. The functional diversity of disome complexes together with RNase digest experiments suggests that paused disome complexes transiently form during ongoing elongation. Structural analysis revealed five disome interfaces between leading and queueing ribosomes that undergo rearrangements as the leading ribosome traverses through the elongation cycle. Our findings reveal at the molecular level how bL9's CTD obstructs the factor binding site of queueing ribosomes to thwart harmful collisions and illustrate how translation dynamics reshape inter-ribosomal contacts.

In all living cells, protein synthesis is orchestrated by the ribosome, which translates the mRNA code into polypeptides that eventually fold into proteins. This cyclical process is organized in four phases: initiation, elongation, termination, and recycling[1]. Driven by large-scale inter- and intra-molecular rearrangements and directed by dedicated protein factors (translation factors), the ribosome traverses through each of the translational phases as it adopts distinct functional states[2]. In vivo, multiple ribosomes translate one mRNA molecule simultaneously, which altogether constitute a polysome. Bacterial polysomes are organized as dense ribosomal clusters with individual ribosomes translating in close proximity, frequently forming higher order assemblies, such as disomes and trisomes[3,4]. In this crowded environment, spatial interference between actively translating ribosomes can have detrimental effects on protein synthesis[5–7]. Thus, an additional layer of regulation is required to prevent traffic jams and harmful ribosome collisions. Using disome complex lifetime as a critical indicator[8], surveillance and quality control pathways distinguish between transient ribosome collisions after which translation can resume[9,10], and irreversibly stalled and collided ribosomes which are targeted for degradation[5–7,11,12].

To study actively translating ribosomes, we previously applied multiparticle cryo-EM to ex vivo-derived human polysomes[13]. While this method allowed us to visualize eleven stable and long-lived intermediates, more transient complexes were lost during polysome

[1]Charité - Univesitätsmedizin Berlin, corporate member of Freie Universität Berlin and Humboldt-Universität zu Berlin, Institute of Medical Physics and Biophysics, Berlin, Germany. [2]Max Planck Institute for Molecular Genetics, Microscopy and Cryo-Electron Microscopy Service Group, Berlin, Germany. [3]Core Facility for Cryo-Electron Microscopy, Charité — Universitätsmedizin Berlin, Corporate Member of Freie Universität Berlin and Humboldt-Universität zu Berlin, Berlin, Germany. [4]Max Delbrück Center for Molecular Medicine in the Helmholtz Association, Technology Platform Cryo-EM, Berlin, Germany. [5]These authors contributed equally: Timo Flügel, Magdalena Schacherl. ✉e-mail: yollete.guillen-schlippe@charite.de; christian.spahn@charite.de

purification. Attempting to study actively translating bacterial ribosomes, we applied the aforementioned approach to the bacterial system, choosing *Escherichia coli* as a model organism. As several intermediates of bacterial translation appeared too transient to be resolved, we amended our approach and reactivated the translational activity of *E. coli* polysomes in the PURE (protein synthesis using recombinant elements) in vitro translation system[14,15]. This method enabled us to visualize the translational landscape of bacterial polysomes during ongoing elongation under defined experimental conditions.

Our data resolve intermediates of all four translational phases, including translation factor bound complexes. Moreover, we obtained structural information on higher-order ribosome assemblies. Strikingly, we find 24 distinct disome complexes composed of eight leading and three queueing 70S ribosome (70S) functional states. We present focused disome interface maps that authentically visualize five specific inter-ribosomal contacts and illustrate elongation-driven structural rearrangements. Structural elucidation of the most populous disome reveals a leading ribosome in a pre-nucleophilic attack state. This state is enriched in the leading disome pool suggesting that slow peptidyl transfer activity leads to translational pauses, which in turn, may cause ribosome collisions and disome formation. Finally, we visualize the key interaction of the mechanism by which bL9 stalls queueing ribosomes upon collision at the sidechain level, thereby complementing recent in situ cryo-ET[4] and cryo-EM[6,7] data.

## Results
### Visualization of bacterial polysomes
In an attempt to visualize actively translating ribosomes on endogenous mRNAs, we purified *E. coli* polysomes ex vivo using a fast gel filtration protocol as described previously[13,16] and subjected them to cryo-EM (Supplementary Fig. 1). In our previous study of actively translating human polysomes we could resolve eleven functional states, including some translation factor bound complexes[13]. From the present data, we obtained reconstructions of eight ribosomal states of bacterial translation (Supplementary Fig. 1e, f).

We found two classical pre-translocational states: one bound to A- and P- tRNAs and one bound to A-, P-, and E-tRNAs. Additionally, we isolated rotated PRE-1 and 2 states with tRNAs in A/P and P/E hybrid positions. Finally, we found two post-translocational (POST) states, one bound to P- and E-tRNAs, while the other bound exclusively to P-tRNA. Thus, we obtained cryo-EM maps for the more stable PRE and POST states, whereas the EF-Tu bound decoding and EF-G bound translocation intermediates are too short-lived to be resolved.

### Polysomal translation activity is restimulated in the PURE system
Aiming for a more comprehensive structural analysis of bacterial translation, we attempted to reactivate the translation activity of bacterial polysomes in vitro, and subsequently image intermediates of active translation by cryo-EM. The principle of reactivation of bacterial polysomes was pioneered in the 1960s and 1970s by Davis and co-workers, who showed that purified *E. coli* polysomes resume translation upon incubation with lysates and S100 extracts[17,18]. Inspired by these early experiments, we employed the PURE in vitro translation system to restimulate the translational activity of ex vivo-derived *E. coli* polysomes under defined controllable conditions.

To assess translational activity, we incubated polysomes as the sole source of ribosomes and mRNA with the PURE system at 37 °C and measured the incorporation of $^{14}$C-labeled Val into nascent chains over a period of 10 min (Fig. 1a). These kinetic measurements demonstrated that ex vivo-derived *E. coli* polysomes indeed resume translation when incubated with aa-tRNAs and translation factors present in the PURE system. After incubation of the polysomes in the PURE system, $^{14}$C-Val incorporation is linear in the initial phase,

followed by a gradual decrease, and eventually plateaus at around 5 min (Fig. 1a). Hence, to visualize polysomes during active translation elongation, we chose a time point during the initial phase and prepared cryo-EM samples of polysomes after 1 min of incubation in the PURE system and subsequently performed multiparticle cryo-EM.

Our structural analysis yielded an almost complete translation cycle, including intermediates of all four phases of translation (Fig. 1b). We found seven elongation intermediates that correspond to previously described states[19–22]. The reactivated polysomes contain three non-rotated PRE states with A-, P-, and E-tRNAs: one bound to partially accommodated A-tRNA, one exhibiting an open small subunit domain conformation, and one in a closed small subunit domain conformation. Moreover, we found rotated PRE-1 and 2 states, a post-translocational (POST) state with P- and E-tRNAs, and a decoding state containing the EF-Tu•GTP•A/T-tRNA ternary complex, as well as P- and E-tRNAs. Notably, we do not find any EF-G bound translocation intermediates.

In addition to the elongation states, the analysis reveals a 30S-like pre-initiation complex (PIC) comprising initiation factor (IF) 1, IF2•fMet•tRNA$^{fMet}$, and IF3, and a termination complex containing a mixture of the peptide chain release factors RF1 and RF2 besides P- and E-tRNAs, resembling the accommodated RF2 bound termination complex[22]. Furthermore, the sample contains a recycling intermediate composed of a 50S ribosomal subunit (50S) and E-tRNA. In line with the kinetic measurements (Fig. 1a), the structural analysis (Fig. 1b) confirms that the PURE system fosters active translation elongation of polysomal ribosomes on the endogenous mRNAs and to some extent, termination, recycling, and initiation.

### Functional disome complexes form dynamically throughout the translation cycle
Upon inspection of each of the isolated 70S classes, we identified extra cryo-EM density adjacent to the bacterial specific ribosomal small subunit protein bS1. To isolate particles showing the extra density, 3D variability was calculated locally for the region around bS1. Subsequently, based on 3D variability clusters, particles were split into two distinct classes. One of the classes showed density for bS1, while the other did not and instead exhibited density for bL9 in a stretched conformation, linking the 70S ribosome to a neighboring 30S ribosomal subunit (30S) (Supplementary Fig. 2b). Suspecting that the neighboring 30S subunit may be part of an entire 70S ribosome, we re-extracted particles of interest at a larger window size and reconstructed them ab initio[23] (Supplementary Fig. 2b). The resulting reconstructions revealed that bL9 in its stretched conformation indeed makes contact with another 70S ribosome that is bound to the same mRNA, resulting in a disome complex (Figs. 1, 2 and 3a).

The present disome complex is in a 'top-to-top' configuration, in which both small subunit heads contact each other. This configuration is consistent with recent cryo-electron tomography (cryo-ET) studies of bacterial polysomes[3,4] and cryo-EM structures of stalled rescue complexes from *E. coli*[6] and *Bacillus subtilis*[7]. Based on these observations, we named the ribosome bound downstream − closer to the 3′ end of the mRNA − the 'leading' ribosome, and the collided ribosome − bound upstream closer to the mRNA's 5′ end − the 'queueing' ribosome. In the following, RNA and protein components of leading and queueing ribosomes are labeled with subscripted letters $_L$ and $_Q$.

The disome complex populations isolated from the reactivated polysome data account for 31% of all imaged 70S ribosomes (Supplementary Fig. 2 and 3). Each of the eight functional 70S state populations contains a fraction of 70 S$_L$ that are part of a disome complex. In comparison to the total 70S functional state distributions, we find no significant enrichment or depletion in 70S$_L$ functional state populations (Fig. 1c).

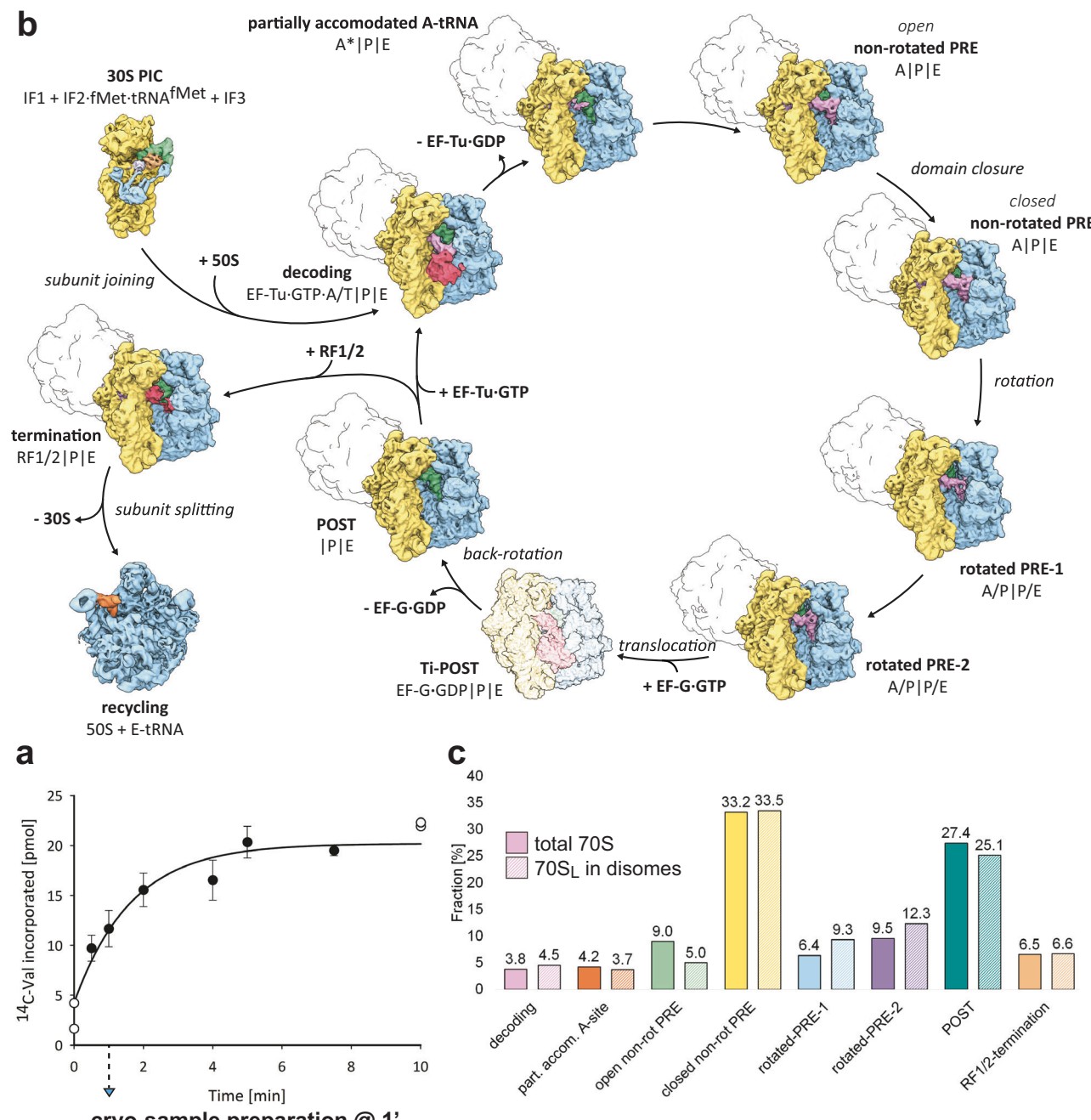

**Fig. 1 | Visualization of actively translating *E. coli* polysomes. a** [14]C-Val incorporation into nascent peptides over a 10-min period after incubation of 700 nM *E. coli* polysomes in the PURE system at 37 °C. Data (closed circles) represent mean values of triplicate experiments with error bars indicating the s. d. (*n* = 3). Data from duplicate experiments are shown individually (open cirles; *t* = 0 and *t* = 10 min). Data were fitted to Eqs. 1 and 2. **b** Translation cycle showing functional ribosome states isolated from ex vivo-derived polysomes reactivated in the PURE system. Shown are 30S (yellow), 50S (blue), initiation factors 1 (lavender), 2 (blue), and 3 (orange), translation factors (red), aminoacyl-tRNAs (A-tRNA, light violet), peptidyl-tRNAs (P-tRNA, green), and exit-tRNAs (E-tRNA, orange). For functional 70S state populations containing disome fractions, collided ribosomes are indicated as silhouettes. The EF-G bound translocation intermediate (Ti-POST: EF-G·GDP | P | E), not observed experimentally, is shown as transparent map simulated from PDB 7N2C[21]. All maps were filtered to 5 Å. **c** Distribution of 70S functional states of all imaged 70S (filled bars) and of 70S$_L$ in disome complexes (dashed bars). Source data are provided as a Source Data file.

Analyzing the initial disome reconstructions, we noticed that intermolecular disome contacts between both 70S ribosomes differed substantially for complexes with non-rotated 70S$_L$ and rotated 70S$_L$ (Fig. 2). In the following, we refer to complexes containing non-rotated 70S$_L$ as 'non-rotated disomes', and to those containing rotated 70S$_L$ as 'rotated disomes'. To investigate whether specific non-rotated or rotated functional states affect the arrangement of the disome interface, we subjected non-rotated and rotated disome particle populations to 3D variability-based classification focused on the interface. Based on the interfaces' 3D variability, we split non-rotated and rotated disome particles into three classes each (Supplementary Fig. 4). Non-rotated classes 1 and 2 show well defined density for leading and queueing 70S, while class 3 shows smeared-out density for the queueing 70S indicating movement of both 70S particles relative to each other (Supplementary Fig. 4a). Across the three non-rotated classes we found neither significant enrichment nor depletion of any leading 70S functional state

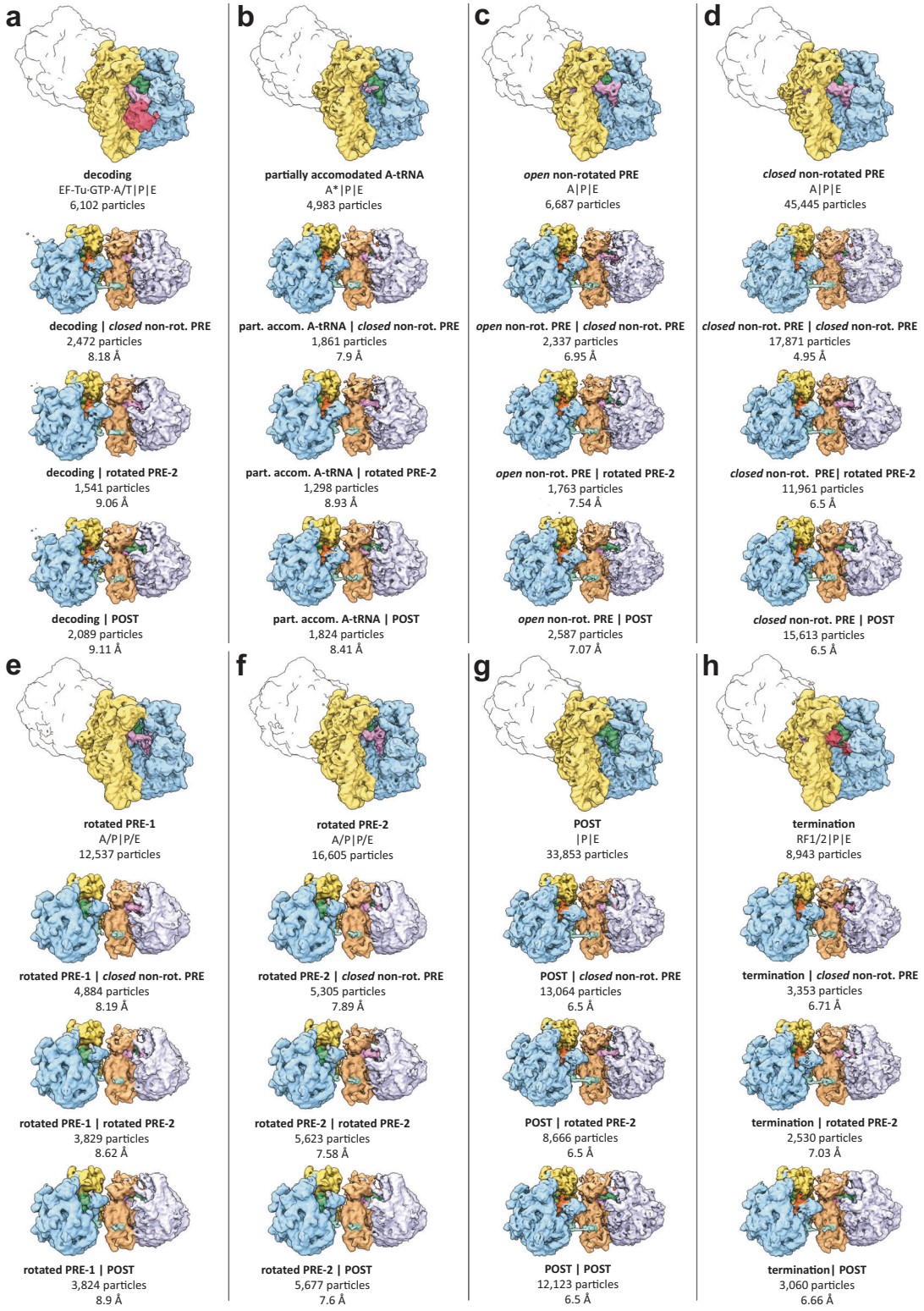

**Fig. 2 | 24 distinct functional disome complexes isolated from reactivated *E. coli* polysomes. a**–**h** For each 70S$_L$ functional state (topmost of each column) three distinct functional 70S$_Q$ states are present: *closed* non-rotated PRE, rotated PRE-2, and POST. Shown are 50S$_L$ (blue), 30S$_L$ (yellow), 50S$_Q$ (lavender), 30S$_Q$ (peach), bL9$_L$ (turquoise), translation factors (red), A-tRNAs (light violet), P-tRNAs (green), and E-tRNAs (orange). Below each disome complex, functional state descriptions of 70S$_L$ and 70S$_Q$ (bold), particle numbers, and reconstruction resolutions are shown.

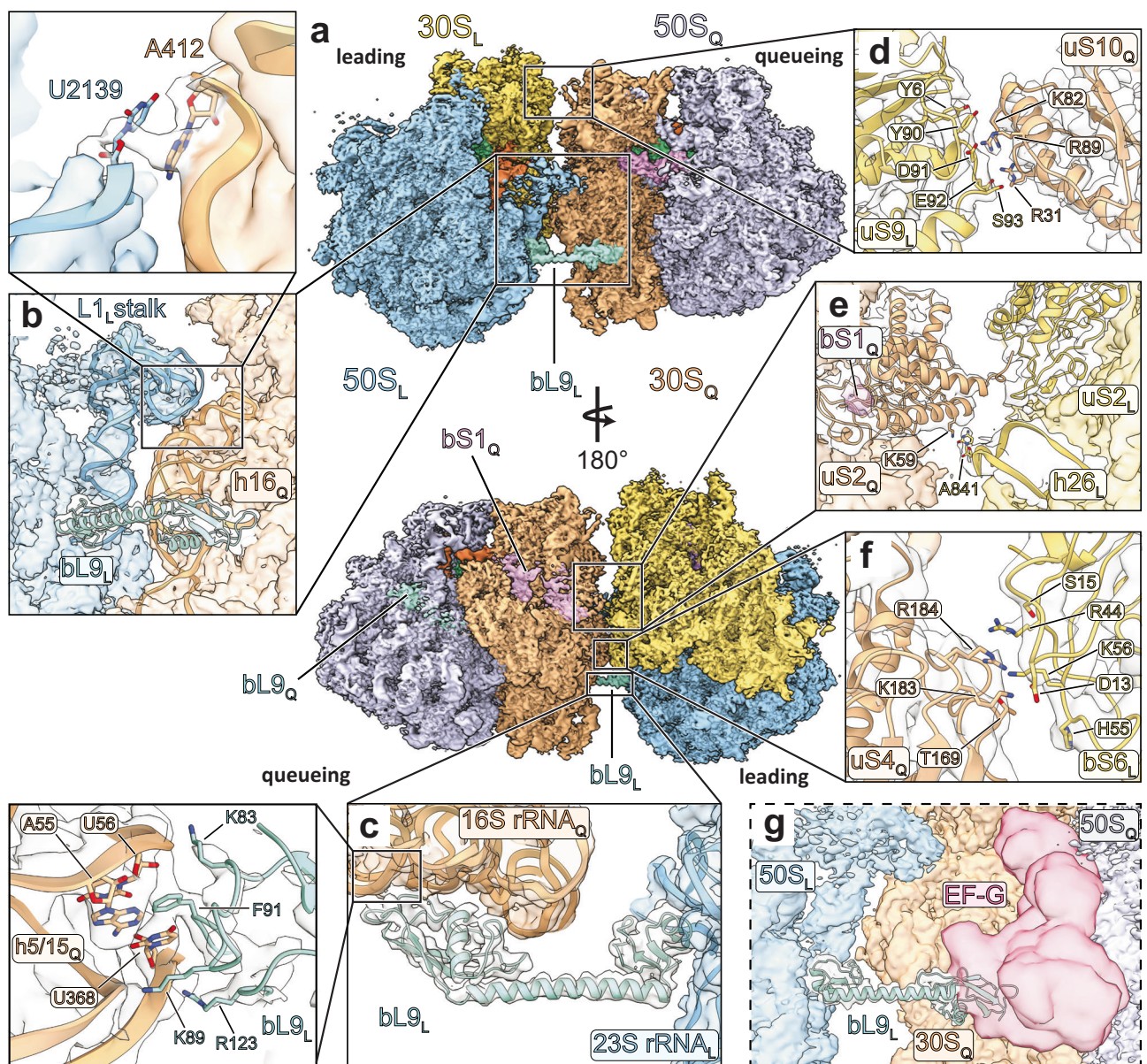

**Fig. 3 | Intermolecular interfaces between 70S_L and 70S_Q. a** Overview of the cryo-EM structure of the disome complex containing 70S_L and 70S_Q in closed non-rotated PRE states. Shown is a composite map generated from locally refined 70S_L, 70S_Q, and interface maps. Shown are 50S_L (blue), 30S_L (yellow), 50S_Q (lavender), 30S_Q (peach), bL9_L (turquoise), EF-G (red), A-tRNAs (light violet), P-tRNAs (green), E-tRNAs (orange), and bS1_Q (pink). Map corresponding to bS1_Q is segmented and shown at lower threshold. Close-ups of the five interfaces between 70S_L and 70S_Q (shown is the interface map). Local resolution ranges of map regions shown in close-ups: (**b**) 4–6.3 Å, (**c**) 3.0–3.6 Å, (**d**) 3.3–3.8 Å, (**e**) 4–4.4 Å, (**f**) 2.7–3.5 Å. **g** Overlay of bL9_L bound to 30S_Q and EF-G (10 Å map simulated from an EF-G•GTP model (PDB: 7N2V[21]).

(Supplementary Fig. 4b). For non-rotated classes 1 and 2, interface distance measurements between 70S_L and 70S_Q indicate no major rearrangements (see interface distances in Supplementary Table 2). Only for the head interface, formed by uS9_L and uS10_Q, we observe a slight opening in non-rotated class 2 relative to class 1. This opening may be correlated to A-site occupation of 70S_Q, as the 70S_Q POST state is more abundant in class 2 compared to class 1 (Supplementary Fig. 4c). Analogously to the non-rotated disome population, classification of the rotated disome interface yielded three distinct classes (Supplementary Fig. 4d). Rotated class 1 shows a higher degree of interface compaction relative to class 2 (Supplementary Table 2). We find an enrichment of the rotated PRE-2 state for 70S_L in class 1, while the rotated PRE-1 state is more abundant for 70S_L in class 2 (Supplementary Fig. 4e). For 70S_Q we find no significant enrichment or depletion of any functional state (Supplementary Fig. 4f).

The high fraction of 31% of all imaged 70S ribosomes arranged as stable disome complexes made us wonder if disome formation critically depends on active protein synthesis. Therefore, we revisited the non-reactivated ex vivo polysome dataset and applied the same classification procedure for disome complexes as described above. Corroborating our previous results, we found a particle population comprising two 70S ribosomes that resemble the disome complex reconstructed from the reactivated polysome dataset (Supplementary Fig. 1g, h). However, this population is significantly less abundant in the non-reactivated dataset, as merely ~8% of 70S particles are arranged as disome complexes. To biochemically explore the relationship between disome complex formation and translational reactivation, we analyzed sucrose density gradient (SDG) profiles of reactivated and non-reactivated polysomes after RNase treatment (Supplementary Fig. 5a, b). Our analysis shows that disomes and trisomes of reactivated

polysomes are more RNase resistant than those of non-reactivated polysomes suggesting that translational restimulation results in an increased number of stabilized disome complexes. To further investigate bL9's role in disome complex stability, we performed the same RNase and SDG experiments for a bL9 deletion strain (Supplementary Fig. 5c). The results show that for both reactivated and non-reactivated polysomes disome and trisome peaks are almost entirely degraded. In contrast, disomes and trisomes of the wildtype show a relatively higher resistance to RNase treatment (Supplementary Fig. 5d). These data indicate that bL9 stabilizes disome and trisome complexes upon RNase digest, potentially due to additional inter-ribosomal contacts.

### $70S_Q$ ribosomes exhibit distinct functional states of translation

To identify the functional states of $70S_Q$, we subjected each of the disome classes from the reactivated polysome dataset to another tier of sorting. Based on 3D variability results for the A-site region, particles were assigned to three distinct classes representing a closed non-rotated PRE state bound to A-, P-, and E-tRNAs, a rotated PRE-2 state with A/P- and P/E-tRNAs in hybrid positions, as well as a POST state bound to P- and E-tRNAs. For all eight functional disome classes (classified based on $70S_L$ functional states), we consistently found these three states for $70S_Q$, respectively (Fig. 2). In total we found 24 distinct disome complexes. These disome reconstructions were refined globally to overall resolutions ranging from 4.95 Å to 9.3 Å. Additionally, we refined the eight $70S_L$ functional states with local masks to resolutions ranging from 3.09 Å to 4.6 Å (Supplementary Fig. 6a–h). Local refinements of the $70S_Q$ states yielded maps ranging from 3.21 Å to 3.3 Å resolution (Supplementary Fig. 6i–k). Upon inspection of the refined intermediates, we found that seven of the $70S_L$ and the three $70S_Q$ states resembled previously described functional states as indicated by cross-resolution calculations (Supplementary Fig. 6a–k). The functional diversity of the disome complexes is remarkable as it shows that 70S in disome complexes adopt distinct functional states of translation, some of which are bound by translation factors (TFs).

### Structure of an elongating disome complex

To achieve a detailed structural analysis of an elongating disome complex, we focused on the most populous state containing a non-rotated closed PRE state as $70S_L$, as well as a non-rotated closed PRE state as $70S_Q$. Local refinements of $70S_L$ and $70S_Q$ resulted in reconstructions of 3.09 Å and 3.3 Å resolution, respectively (Supplementary Fig. 6d, i, Supplementary Fig. 7 a,b,d, e).

To better visualize the contacts between both ribosomes, non-rotated disome particles (referring to non-rotated $70S_L$) − disregarding the exact functional state − were merged as they exhibited a homogeneous interface. These particles were refined with a mask surrounding both small subunits, the $L1_L$ stalk and $bL9_L$, yielding a disome interface map of 3.28 Å resolution (Supplementary Fig. 6c). These three maps were then used for atomic model building and subsequent disome complex interpretation.

Five specific interfaces between $70S_L$ and $70S_Q$ define the disome complex. The most prominent interaction is formed by the stretched $bL9_L$ and the $30S_Q$ (Fig. 3a, b). Through its C-terminal domain (CTD), $bL9_L$ makes contacts with the 16S $rRNA_Q$ helices h5, h15, h16, and h17. As initially described by Xue et al. at intermediate resolution[4], this interaction anchors $bL9_L$ to $70S_Q$ and simultaneously obstructs the factor binding site of $70S_Q$ (Fig. 3c). The notion that $bL9_L$ binding to $70S_Q$ induces a tight disome arrangement is corroborated by our RNase digest experiments that show that bL9 crucially stabilizes disome complexes (Supplementary Fig. 5). Due to the obstruction of the $70S_Q$ factor binding site, binding of the elongation factors EF-Tu and EF-G is forestalled (Fig. 3g). $70S_Q$ are therefore locked in PRE or POST states, respectively. Three residues at the tip of the $bL9_L$ CTD are directly involved in this interaction (Fig. 3c): Phe91 reaches into the

groove of $h5_Q/h15_Q$ and stacks with G57, as well as with U368, which, in turn, base-pairs with nearby A55. On the opposite side of U368, Lys89 seems to contact the base potentially through a stacking interaction. Furthermore, Lys83 contacts the phosphate backbone between G57 and C58. In contrast, $bL9_Q$ is found in the compact conformation with its CTD bound to $uL2_Q$, $bS6_Q$, and 23S $rRNA_Q$ H79 (Fig. 3a). Another major contact is formed between the $L1_L$ stalk and the $30S_Q$ body, involving 23S $rRNA_L$ H78 and 16S $rRNA_Q$ h16 (Fig. 3b). Here, U2139 (23S $rRNA_L$) stacks with A412 (16S $rRNA_Q$), reinforcing the interaction.

$30S_L$ and $30S_Q$ constitute three additional interfaces. Between both 30S heads, one interface is formed through electrostatic and polar sidechain interactions of $uS9_L$ and $uS10_Q$ (Fig. 3d). $uS2_Q$ establishes another interface through interactions with $uS2_L$ and $h26_L$ on the side opposite to the $L1_L$ stalk (Fig. 3e). Further down the $30S_Q$ body, $bS6_L$ and $uS4_Q$ form an additional interface involving charged and polar sidechain interactions (Fig. 3f). Taken together, our structures reveal five specific interfaces between $70S_L$ and $70S_Q$ that define a functional disome complex.

Interestingly, while $bS1_Q$ is present in the disome complex, $bS1_L$ is not resolved (Fig. 3a). As described above, this difference has initially guided us during image classification. It is noteworthy that in this specific disome configuration, $bS1_L$ would sterically clash with $70S_Q$ (Supplementary Fig. 8a, b). Therefore, $bS1_L$ may either dissociate from $70S_L$ or adopt a flexible conformation outside of the interface. To further investigate $bS1_Q$ in the context of the present disome complex, we low-pass filtered the $70S_Q$ map and docked an AlphaFold2[24] generated model of bS1 into it (Supplementary Fig. 9a). The filtered density readily accommodates the two N-terminal bS1 domains and extends further towards the putative location of bS1 domain 3.

### Pre-attack state is enriched in $70S_L$

Upon inspection of the most populous disome state formed by non-rotated closed PRE states with tRNAs in classical A-, P-, and E-sites for $70S_L$, as well as for $70S_Q$, we initially assumed both particles to be in a post-nucleophilic attack state. However, close analysis of cryo-EM maps and respective atomic models reveals an intriguing difference in the peptidyl transfer center (PTC) region. In line with the corresponding population of total 70S (Supplementary Fig. 2b, closed non-rotated PRE), $70S_Q$ is in a post-attack state after peptide bond formation with peptidyl-tRNA in the A-site and deacylated tRNA in the P-site (Fig. 4a). In contrast, $70S_L$ appears to be in a pre-nucleophilic attack (pre-attack) state, in which the nascent chain has not yet been transferred from the P-site tRNA to the aminoacyl-tRNA in the A-site (Fig. 4b). Notably, the positioning of the CCA-ends as well as the key 23S rRNA residues involved in peptidyl transfer (G2061, A2062, and U2506) are consistent with the available high resolution cryo-EM and crystal structures of reconstituted pre-attack states (PDBs: 6WDD[20] and 8CVJ[25]). These results indicate an enrichment of the pre-attack state in $70S_L$. It is possible that impaired or slow peptidyl transfer activity of the $70S_L$ may result in a pausing event, and subsequently in a collision with a $70S_Q$. The non-rotated closed PRE pre-attack state accounts for 33% of all $70S_L$ (Fig. 1c), thereby representing the most abundant $70S_L$ state. Thus, tardy peptidyl transfer activity appears to be a significant cause of ribosome pauses, at least in our experimental system.

### Higher order organization of *E. coli* polysomes

Throughout the cryo-EM dataset we observed dense polysomal clusters in electron micrographs (Supplementary Fig. 2a). Therefore, we asked if our data would resolve higher order polysomal arrangements, such as trisome complexes. To address this question, we merged and aligned all non-rotated disomes and rotated disomes, respectively. 3D variability-based classification resulted in two distinct particle populations, one of which represents a disome,

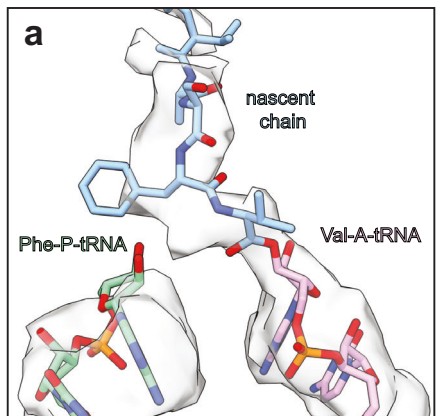
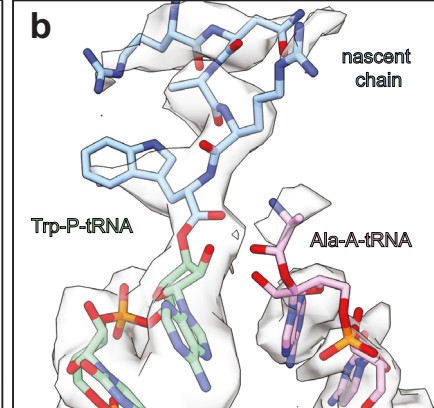

**Fig. 4 | The pre-nucleophilic attack state is enriched in 70S$_L$.** Close-ups of the peptidyl transferase centers (PTC) of 70S$_Q$ (**a**) and 70S$_L$ (**b**). The cryo-EM densities reflect the weighted average of all tRNA species and amino acid occupants. **a** 70S$_Q$ bound to Val-A-tRNA (light violet), Phe-P-tRNA (green), and nascent peptide chain (blue) is in a post-nucleophilic attack state in which the nascent peptide has been transferred to the A-site tRNA. **b** 70S$_L$ bound to Ala-A-tRNA (light violet), Trp-P-tRNA (green), and nascent peptide chain (blue) represents a pre-nucleophilic attack state in which the nascent peptide has not yet been transferred to the A-site tRNA.

the other one a trisome complex. For non-rotated and rotated populations, we found trisome fractions accounting for 11% and 38% of the respective disome classes. Overall, trisomes make up 8.1% of all imaged 70S particles.

The non-rotated and rotated trisome complex populations were refined globally to resolutions of 7.6 Å and 7.8 Å, respectively (Supplementary Fig. 9c, e). Analysis of these reconstructions revealed trisome complexes in a 'top-to-top-to-top' ('t-t-t') configuration which is established through the same five intermolecular interfaces described above (Supplementary Fig. 9b, d). This arrangement is in line with in situ cryo-ET studies of polysomes in *M. pneumoniae*[4]. The present trisome maps represent a 70S$_L$ in complex with two 70S$_Q$, which we termed 70S$_{Q1}$ and 70S$_{Q2}$. Interestingly, while the factor binding site of 70S$_{Q1}$ is blocked by the bL9$_L$ CTD, bL9$_{Q1}$ is in a stretched conformation, binding with its CTD to the factor binding site of 70S$_{Q2}$ (Supplementary Fig. 9b, d). This interaction distinguishes 70S$_{Q1}$ from 70S$_Q$ in the disome arrangement. In contrast, bL9$_{Q2}$ is in a compact conformation. Another striking difference between 70S$_{Q1}$ and 70S$_{Q2}$ is the absence of bS1$_{Q1}$ density, while density for bS1$_{Q2}$ is present.

### Intersubunit-rotation of 70S$_L$ induces L1$_L$ stalk dissociation from 30S$_Q$

To gain further insights into disome interface dynamics, we investigated how disome contacts change as 70S$_L$ traverse through the elongation cycle. Remarkably, upon intersubunit-rotation of 70S$_L$, the L1$_L$ stalk dissociates from the 30S$_Q$, thereby dissolving one of the five intermolecular interfaces (Fig. 5a). The intersubunit-rotation is classically defined as a rotation of the small ribosomal subunit relative to the large ribosomal subunit. However, in the context of the disome, 50S$_L$ undergoes intersubunit rotation, while 30S$_L$ is kept in place by interactions with 70S$_Q$. In turn, the aforementioned disome configuration is visibly disrupted (Supplementary Movie 1). During this structural rearrangement, bL9$_L$ remains anchored to the factor binding site of 70S$_Q$ through its CTD. However, concomitant with L1$_L$ stalk dissociation, density for bL9$_L$'s linker helix becomes fragmented indicating increased flexibility.

To further investigate the structural rearrangements induced by 70S$_L$ intersubunit rotation, we refined the rotated interface classes 1 and 2 to resolutions of 4.46 Å and 5.32 Å, respectively (Supplementary Fig. 6g, h). Inspection of these maps shows that in the rotated interface, apart from the L1 stalk contact, the uS4$_Q$:bS6$_L$ contact dissolves, while another interface between uS11$_L$ and uS4$_Q$ is formed (Fig. 5b–d). Rigid body docking of the bL9 CTD into rotated classes 1 and 2 maps while aligning both structures onto the 16S rRNA$_Q$ revealed an 8.7 Å

movement of the CTD along the h5/15 groove (Fig. 5f). Notably, in the rotated class 2 conformation the sidechain of Phe91 would be inserted less deep into the h5/15 groove likely resulting in a weaker interaction of bL9$_L$ to 70S$_Q$. Analogously, analysis of the non-rotated interface classes 1 and 2 (Supplementary Fig. 4a) showed no significant variation in the positioning of the bL9 CTD bound to h5/15 (Fig. 5g).

## Discussion

Using a combined approach of translational reactivation of ex vivo *E. coli* polysomes in the PURE system and multiparticle cryo-EM, we were able to visualize distinct functional intermediates of all four phases of translation during ongoing elongation. This method allowed us to resolve transient translation factor-bound intermediates, such as TC-bound decoding and RF1/2-bound termination complexes. We find that all 70S states isolated from our data are also present as functional disome complexes that contain bL9$_L$ in its stretched conformation. Through focused classification and refinement of the disome interface we were able to resolve five disome interfaces and visualize how these interfaces are reorganized as 70S$_L$ traverses through the elongation cycle. bL9 plays a key role in the organization of the interface. Through its interaction with 70S$_Q$, bL9$_L$ acts as a spacer between two 70S upon collision and concomitantly blocks recruitment of TFs to 70S$_Q$[4,6,7]. Our non-rotated disome interface structure illustrates how residues of the bL9$_L$ CTD specifically interact with the 16S rRNA$_Q$ to block elongation factor binding to 70S$_Q$.

The overall structure of the functional disomes is reminiscent of the recently published disome rescue complexes from *E. coli*[6] and *B. subtilis*[7]. While these complexes were stably stalled by VemP and chloramphenicol and subsequently biochemically enriched, our complexes form transiently, most likely due to pausing of a leading ribosome rather than a permanent stalling event. The disome rescue complexes show bL9 in a conformation consistent with our structure, albeit in a different functional context. These functional differences are highlighted e.g., by the missing uS2$_L$ in the case of *B. subtilis* disome. The absence of uS2$_L$ may allow the *B. subtilis* disome to adopt a tighter interface compared to the *E. coli* disome[6]. Unlike in the case of the chloramphenicol-stalled *B. subtilis* disome, in which bL9 locks the collided ribosome in a rotated state[7], our data show that under unperturbed conditions, queuing ribosomes adopt three distinct functional states: classical PRE, rotated PRE-2, and POST (Fig. 2). Comparing our structure with the VemP-stalled and SmrB bound *E. coli* disome, we find that the bS1$_Q$ NTD of the functional disome occupies the proposed binding site of the SmrB N-terminal helix (Supplementary Fig. 8c). The occupation of the SmrB binding site by bS1$_Q$

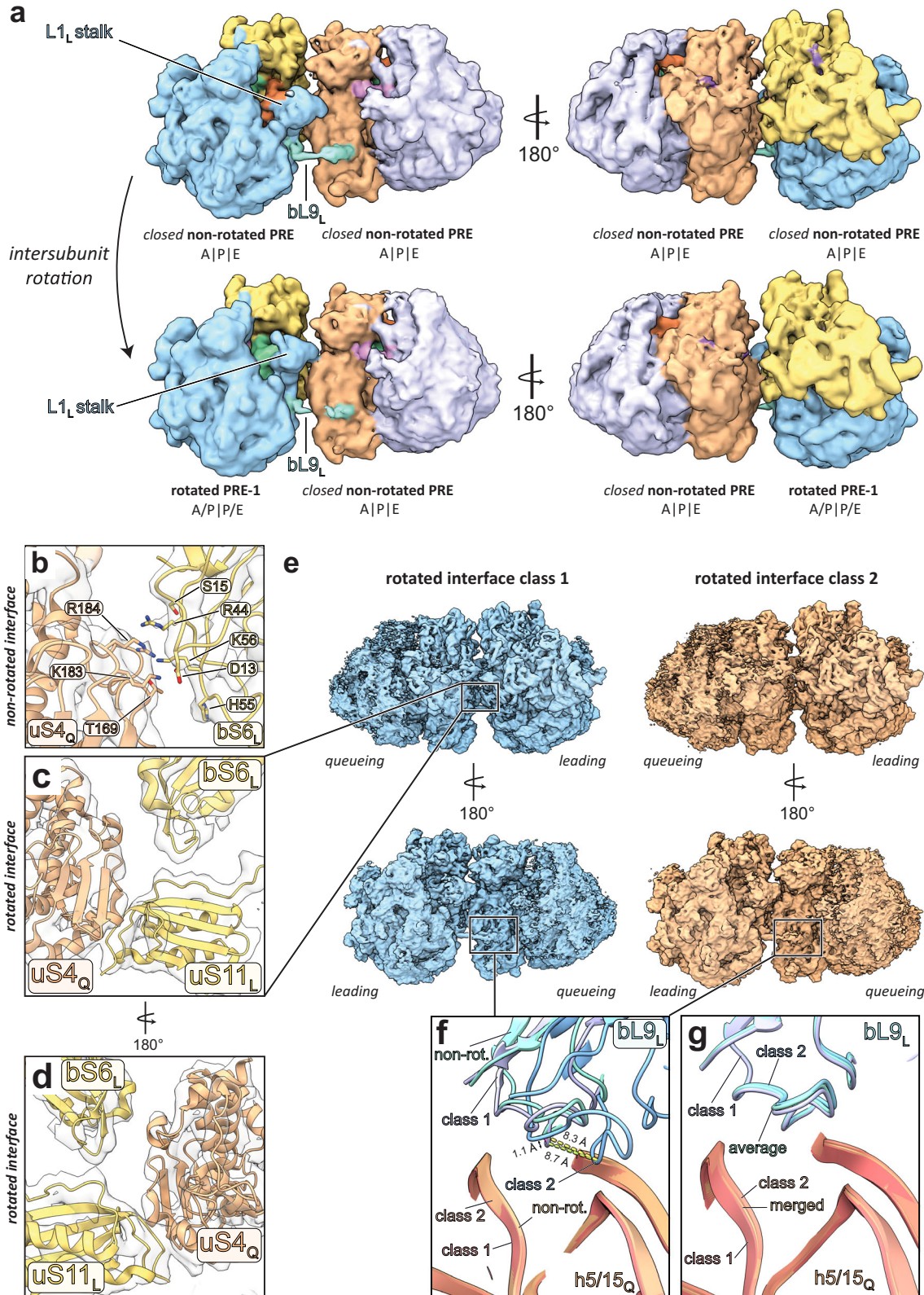

**Fig. 5 | Disome interface dynamics during 70S_L elongation. a** Intersubunit rotation of 70S_L induces L1_L stalk dissociation from 30S_Q while bL9_L CTD remains bound. Shown are 50S_L (blue), 30S_L (yellow), 50S_Q (lavender), 30S_Q (peach), bL9_L (turquoise), A-tRNAs (light violet), P-tRNAs (green), E-tRNAs (orange), and mRNA (purple). **b**–**d** non-rotated bS6_L:uS4_Q interface is dissolved upon 70S_L intersubunit rotation and a new interface between uS11 and uS4_Q forms. **e** Cryo-EM maps of

rotated interface classes 1 (blue) and 2 (orange). **f** Overlay of rotated interface classes 1 and 2 with non-rotated disome (PDB: 8R3V) interface model showing an 8.7 Å upward movement of the bL9_L CTD along h5/15. **g** Overlay of non-rotated interface classes 1 and 2 with the average non-rotated (PDB: 8R3V) disome interface model showing no significant conformational differences of the bL9_L CTD.

structurally distinguishes the paused disome from the stalled rescue complex. The formation of paused disome complexes may be necessary but not sufficient for SmrB to bind. Additional rescue factors may be required to recognize irreversibly stalled disome complexes and to induce the necessary structural rearrangements that allow for SmrB— and potentially other endonucleases involved in the rescue pathway— to bind to the disome interface. Alternatively, irreversibly stalled and paused disome complexes may be discriminated by a kinetic mechanism with the lifetime of the disome complex as a critical parameter.

It is remarkable that bS1 is only resolved in $70S_Q$ and $70S_{Q2}$ (Fig. 3a and Supplementary Fig. 8) but not in $70S_L$ or $70S_{Q1}$, suggesting that bS1 marks the last ribosome in the queue. bS1's functional role during polysomal elongation may be to protect upstream mRNA. However, it remains elusive why bS1 is neither visible in $70S_L$ nor $70S_{Q1}$. bS1 may either dissociate from $70S_L$ and $70S_{Q1}$ or adopt a flexible conformation outside the disome and trisome interfaces upon contact with upstream elongating 70S. It is noteworthy that in the present trisome structure the space outside the interfaces is constricted by the tight arrangement of $30S_L$, $30S_{Q1}$, and $30S_{Q2}$ (Supplementary Fig. 9b–e), making it difficult to accommodate $bS1_L$ and $bS1_{Q1}$. Additionally, superimposition of $bS1_Q$ on $70S_L$ shows a steric clash with $70S_Q$ (Supplementary Fig. 8a, b), suggesting that in the present complex $bS1_L$ cannot occupy its canonical binding site on the small subunit.

Our structural data suggest that bacterial ribosomes have a significant propensity to oligomerize during ongoing translation and that disome complexes can readily form in diverse functional contexts. Recent studies have investigated the various physiological conditions that slow down elongation[26–31], cause translational pauses[32–39], and, if pauses are prolonged, eventually lead to disome complex formation[10,35,39–42]. In principle, frequent collisions between $70S_L$ and $70S_Q$ may arise from temporary differences in elongation speed— either of stochastic or site-specific nature due to e.g., rare codons. Considering the reported in vivo elongation speed of ~20 aa s$^{-1}$[3], a translational pause of one second would suffice for $70S_Q$ to shorten the distance to $70S_L$ by 60 nt, which is about the size of an *E. coli* disome footprint (57 nt)[35]. Given that in vivo studies found that ~82% of 1038 examined *E. coli* genes are translated with one or more events of pausing[36], pause-induced formation of transient disome complexes is likely to be a frequent event in bacterial cells.

It is noteworthy that we find significantly more disome complexes for the re-stimulated polysomes (31%) than for the non-stimulated polysomes (7.7%). Thus, it may be possible that suboptimal translation conditions in our system led to enhanced disome formation. Compared to the physiological environment, the PURE system uses a higher $Mg^{2+}$ concentration, lacks additional factors such as EF-P, and uses unmethylated RFs. It has been shown that high $Mg^{2+}$ concentrations slow down E-tRNA release, which in turn, decreases elongation speed[26]. Moreover, the lack of EF-P, which increases the reactivity of poor aminoacyl subtrates[43,44], may explain to some extent the high abundances of disomes containing $70S_L$ in the pre-attack state. Finally, the usage of unmethylated RFs might result in slow termination[45,46] followed by ribosome queues at stop codons. However, the frequent formation of disome complexes under conditions of active translation may actually be reflective of the spatial organization of polysomes in a bacterial cell. This notion is supported by in situ cryo-ET studies of *M. pneumoniae* that found that 33% of polysomal 70S particles contain bL9 in its stretched conformation[4].

The fact that the non-rotated pre-attack state is the most abundant $70S_L$ state (33.5%) suggests that slow peptidyl transfer is a common cause of translational pauses and disome formation in our system. This observation is in line with integrated nascent chain profiling (iNP) data that showed a high frequency of puromycin resistant pauses in *E. coli* indicating halted peptidyl transfer reactions[35,36]. As the present disome complexes contain endogenous mRNA, we expected a mixture of tRNAs to be bound to A-, P-, and E-sites, respectively. Accordingly, the observed tRNA cryo-EM densities reflect the weighted average of all tRNA species and amino acid occupants. In line with this notion, we observe cryo-EM density that may accommodate various amino acids in the non-rotated pre-attack $70S_L$ state. However, a specific context may have been enriched for $70S_L$ in the non-rotated pre-attack state, as slow peptide bond formation appears to depend on the nature of the involved amino acids[47]. Interestingly, iNP data by Fujita et al. suggest an enrichment of Ala-tRNA for the A-site and Trp-tRNA for the P-site in $70S_L$ in the context of disome pauses[35]. Guided by these data, we built atomic models for an Ala-tRNA for the A-site and a Trp-tRNA for the P-site (Fig. 4a, b).

Previous studies of collided ribosomes provided insights into the intermolecular organization of bacterial disome complexes[6,7]. These analyses were based on composite structures generated from individually refined $70S_L$ and $70S_Q$ maps. In contrast, we obtained five focused maps of non-rotated and rotated disome interfaces classes. This enabled us to authentically visualize the contacts between $70S_L$ and $70S_Q$ and investigate the conformational changes they undergo during elongation. Our interface model provides mechanistic insights into the bL9 CTD interaction with the 16S $rRNA_Q$; $bL9_L$'s key interacting residue is Phe91, which reaches into the groove of $h5/15_Q$ and stacks with U368, which in turn base pairs with A55. This interaction is stabilized by the positively charged residues Lys83 and Lys89. Various crystal structures of the *E. coli* 70 S show an overall similar arrangement in the crystal unit cell but differ in the exact binding modes for the bL9 CTD to 16 S rRNA of neighboring ribosomes, due to crystal packing. In all structures we analyzed (PDBs: 4YBB[48], 4V4Q[49], 6BY1[50]), the interacting bL9 residues are the same: Lys83, Lys89, and Phe91. However, alignment of the crystal structures with our (non-rotated) disome structure revealed a linear trajectory of binding modes that reaches from the lower end of the h5/15 groove to the top end spanning a distance of 27 Å (Supplementary Fig. 10). Our structure comparison illustrates the high plasticity of bL9 CTD binding, which allows for a certain degree of movement along the groove of h5/15. This plasticity may permit $bL9_L$ to remain bound to $70S_Q$ during large scale movements in the disome interface. In line with this, we observed that upon $70S_L$ intersubunit-rotation $bL9_L$'s CTD moves 8.7 Å along the groove of h5/15 (Fig. 5f). In this conformation the tip of the CTD containing the residues Lys83-Phe91 closely overlaps with the *E. coli* 70S crystal structure by Noeske et al. (PDB: 4YBB)[48] (Supplementary Fig. 10h). Upon the rotation induced repositioning, the sidechain of Phe91 cannot reach as deep into the h5/15 groove as in the non-rotated interface conformation (Fig. 3c, S9c). Consequently, the rotated interface class 2 likely exhibits a weaker interaction between $bL9_L$ CTD and 16S $rRNA_Q$.

The intrinsically tight arrangement of bacterial polysomes[3,4] and the various scenarios in which ribosomes pause[26,32–39] and queue during elongation necessitates a productive mechanism by which harmful collisions are thwarted and frameshifting followed by aberrant protein production are forestalled. As shown by our RNase digest experiments, bL9 crucially stabilizes elongating disome complexes, while in its absence disome and trisome peaks are degraded (Supplementary Fig. 5). We therefore hypothesize that the stable disome configuration readily dissolves upon $bL9_L$ dissociation from the factor binding site of 70 $S_Q$. It is conceivable that the conformational changes induced by $70S_L$ intersubunit-rotation (Fig. 5) and thereupon $70S_L$ translocation events facilitate $bL9_L$ dissociation. In line with our structural and biochemical analyses and recent in situ cryo-ET studies of *M. pneumoniae*[4], we suggest a bL9 driven mechanism of dynamic disome formation during active translation elongation (Fig. 6). Through the action of bL9, transiently forming disome complexes may act as safeguards to prevent frame shifting and aberrant protein synthesis, allowing for translation to resume after ribosome collisions.

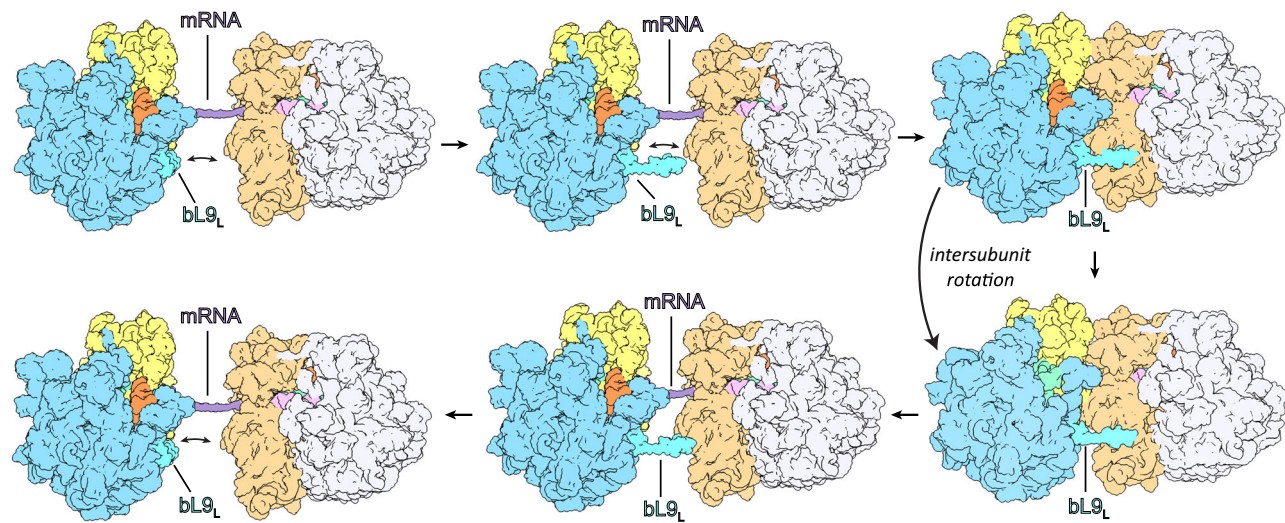

**Fig. 6 | Putative mechanism of functional disome formation and dissociation.**
Two individual 70 S translate the same mRNA. $bL9_L$ switches between compact and stretched conformation, sampling the region upstream the mRNA exit. Upon translational pausing of $70S_L$, $70S_L$ and $70S_Q$ collide. The direct contact between $70S_L$ and $70S_Q$ stabilizes $bL9_L$'s stretched conformation, induces binding of its CTD to $30S_Q$, and the establishment of four additional interfaces. $bL9_L$ CTD binding stalls $70S_Q$ through obstruction of its factor binding site. Intersubunit rotation of $70S_L$ induces $L1_L$ stalk dissociation from $30S_Q$ while $bL9_L$ remains bound. Upon translocation events of $70S_L$, $bL9_L$ dissociates. $70S_Q$ is no longer stalled and both $70S_L$ and $70S_Q$ resume translation as $bL9_L$ returns to the sampling mode. Shown are $50S_L$ (blue), $30S_L$ (yellow), $50S_Q$ (lavender), $30S_Q$ (peach), $bL9_L$ (turquoise), A-tRNAs (light violet), P-tRNAs (green), E-tRNAs (orange), and mRNA (purple).

The diverse set of disome complexes presented here demonstrates that disomes form transiently during ongoing translation and that 70S in a disome arrangement can adopt distinct functional states, including TF bound states. The focused interface maps provide structural insights into organization of inter-ribosomal contacts including the mechanism by which $bL9_L$ binds to $70S_Q$. Recent studies showed that mutations and deletion of bL9 result in frame shifting[51,52]. Our data further elucidate the structural mechanism of how bL9 performs its proposed frame-keeping function to thwart harmful collisions during translation elongation. Finally, our structures illustrate that the disome interface, governed by elongation dynamics, is reshaped while the disome complex is maintained throughout POST and PRE states. Together, these results indicate that disome complexes play an active role in bacterial translation.

## Methods

### Preparation of ex vivo *E. coli* polysomes

Intact *E. coli* polysomes assembled on their native mRNA substrates were isolated from cytoplasmic extracts of bacterial cultures growing in exponential phase, by rapid extraction and size exclusion chromatography, adopted from our in-house protocol of mammalian polysomes[13]. The MRE600 strain was chosen due to its low RNAse I activity, which has established this strain as a reliable source for the purification of intact RNA and ribosomes from *E. coli*[53]. All bacterial cultures and preparative steps were performed in the absence of antibiotics or inhibitors. Standard precautions regarding laboratory equipment and solutions were taken to prevent nucleolytic degradation of polysomes and ribosomes, and all preparative steps and centrifugations were performed fast and in the cold. For bacterial cultures and the preparation of the cytoplasmic extracts, we adopted a published protocol[54] and performed the following steps: Bacterial cultures were grown in LB medium at 37 °C until an absorbance of 0.5 OU/mL at 600 nm. The cell pellet was harvested by centrifugation at $5000 \times g$ for 20 min and was resuspended in 2 mL buffer T (20 mM HEPES·KOH pH 7.6, 6 mM Mg-Acetate, 30 mM K-Acetate, 2 mM DTT). The cells were broken by the addition of 0.4 mg Lysozyme (Fluka, product number # 62970) per mL cell suspension, followed by two freeze-thaw cycles. The resulting, very viscous, lysate was treated with 10 Units DNAse I

(RNAse free, NEB #M0303S) per mL suspension, for 10 min on ice followed by short centrifugation through QIA shredders tubes, and the clear cytoplasmic extract was separated from cell debris by centrifugation at $12,000 \times g$ for 6 min. The extract was concentrated and applied to an equilibrated Sepharose 4B column of 24 mL in buffer GF (50 mM HEPES·KOH pH 7.6, 6 mM Mg-Acetate, 100 mM K-Glutamate, 2 mM Spermidine, 0.05 mM Spermine, 1 mM DTT). Polysomes were collected from the fractions corresponding to the first peak with absorbance at 254 nm; these were concentrated and either used directly for grid preparation or were flash frozen in 5 µl aliquots for in vitro translation reactions. The quality of the preparation was determined by sucrose gradient centrifugation, RNA gel electrophoresis and electron microscopy (Supplementary Fig. 1). For analytical polysome profiling by ultracentrifugation, 2 OU (48 pmol 70 S) of the final preparation was separated by centrifugation for 15.5 h at $24,806 \times g$ in 10–45% sucrose gradients, prepared in buffer GF, using a SW40 rotor. For rRNA analysis, total RNA of lysate and polysome aliquots were prepared with TRIZOL and separated on a 1% formaldehyde agarose gel, before being stained with SYBR Green II.

### Kinetic translation assays

Translations were performed in the PURE translation system using the PURExpress delta ribosome kit (NEB, #E3313S) according to the manufacturer's protocol with a few modifications. Translation reactions were supplemented with 0.8 U/µL RNAsin Plus RNase Inhibitor (Promega, N261B). SolA, factor mix, 0.014 µCi/µL $^{14}$C-Val (Perkin Elmer, NEC291E, 271 µCi/µmol, 0.05 µCi/µL) and RNAsin Plus were combined on ice, followed by a preincubation at 37 °C for 2 min, and added directly to polysomes (0.7 µM final concentration) that had been preincubated at 37 °C for 2 min. At the indicated time points 10 µL of the reaction mixture was withdrawn and quenched with 10 µL 2 M NaOH, followed by a 10 min incubation at 37 °C. The samples were neutralized with 10 µL of 2 M HCl, combined with 30 µL 1% BSA and polypeptides precipitated with 1 mL ice cold 10% TCA. Precipitated material was collected on glass filters (Whatman GF6, 25 mm, 10370018), washed 3 times with 5% ice cold TCA, followed by 3 washes with 100% Ethanol. Filters were dried for 1 h at 60 °C, combined with universal liquid scintillation cocktail (0016.3; Rotiszint) and the amount of $^{14}$C-Val

incorporated into polypeptides determined using a liquid scintillation counter (Wallac 1409). Each data point was obtained at least in duplicate. Control reactions without polysomes were performed to set the background. The specific activity of valine in the reaction was determined by dividing the total counts by the total amount of valine present in the reaction mix ($^{14}$C-labled and unlabeled valine), under the assumption that the PURExpress delta ribosome kit contains 300 μM valine. The amount of pmol valine incorporated was calculated by dividing the TCA precipitated counts by the measured specific activity of the reaction mix. To calculate the rate of protein synthesis, initial rate of of 8,4 pmol Val/min was determined by fitting the data up to 1 min to $y = y0 + a*x$ (1), and divided by the fraction of valine content in the newly synthesized polypeptides, which was based on the estimate of approximately 0.055, to obtain 2.5 pmol amino acid incorporation per second[55]. The estimated amount of 7 pmol/μL of ribosomes present in the translation reactions represents an upper limit and the calculated protein synthesis rate of 0.36 aa/sec/ribosome a lower limit for the calculated rate. Data were plotted using SigmaPlot 14 and the data fit to a single exponential rise to a maximum with 3 parameters:

$$y = y0 + a*[1 - \exp(-b*x)] \qquad (2)$$

## Bacterial strains and plasmids

*E. coli* polysomes used for all biochemical and structural experiments were isolated from the MRE600 strain. *E. coli* wild-type (BW25113) and ΔbL9 (BW25113ΔrplI::kan/JW4161) strains were obtained from Horizon Discovery. The kanamycin cassette was deleted from the ΔbL9 strain with FLP recombination using the temperature-sensitive pCP20 plasmid (NovoPro Bioscience Inc.)[56,57] (Barrick, J. E. FLP recombination in *E. coli*. https://barricklab.org/twiki/bin/view/Lab/ProcedureFLPFRTRecombination). Briefly, electrocompetent JW4161 cells were electroporated with the pCP20 plasmid and transformants grown at 30 °C overnight on Luria broth (LB) /ampicillin plates to allow for recombination and excision of the kanamycin cassette. Single colonies were inoculated in LB media and grown overnight at 43 °C to select for loss of pCP20. The overnight culture was plated on LB plates and grown at 30 °C overnight. Single colonies were screened for genomic recombination, by replating single colonies in the following order: LB/kanamycin, followed by LB/ampicillin, and then on LB plates and grown overnight at 30 °C. Kanamycin and ampicillin sensitive colonies were chosen as successful recombinants not containing the kanamycin cassette or the pCP20 plasmid. Polysomes were isolated from the wild-type (BW25113) and ΔbL9 (BW25113ΔrplI/JW4161) strains, as described above for the MRE600 strain, and also used for the nuclease digest experiments.

## Nuclease digestion

In vitro translation reactions were performed in the PURE translation system using the PURExpress delta ribosome kit (NEB, #E3313S) as described above. Translation reactions were supplemented with 0.8 U/μL RNAsin Plus RNase Inhibitor (Promega, N261B). SolA, factor mix, and RNAsin Plus were combined on ice, followed by a preincubation at 37 °C for 2 min, and added directly to polysomes (0.7 μM final concentration) that had been preincubated at 37 °C for 2 min. No translation samples were prepared identically, with the exception that factor mix was omitted from the reactions. After 1 min reaction time, translation reactions were quenched by diluting the translation reaction 5-fold with translation stop buffer (20 mM Tris pH 8 150 mM MgCl$_2$ 0,1 M NH$_4$Cl 5 mM CaCl$_2$) and layered over an equal volume sucrose cushion (1.1 M Sucrose 20 mM Tris pH 8 500 mM NH$_4$Cl 10 mM MgCl$_2$, 0.5 mM EDTA pH 8). Ribosomes were pelleted by centrifugation using a TLA 110 rotor at 82,575 g for 3 h at 4 °C. Pellets were quickly washed with ice cold resuspension buffer (20 mM Tris pH 8 10 mM

MgCl$_2$ 100 mM NH$_4$Cl, 5 mM CaCl$_2$) and dissolved in the same buffer. 14 pmols of polysomes (based on A260) were combined with RNAseA (final concentration of 0.25 μg/mL Thermo Fisher Scientific, EN0531) for 1 h at 37 °C, quenched with 25 U SuperaseIn (Invitrogen, AM2696), diluted to 100 μL in GF buffer (50 mM HEPES-KOH pH 7.6, 6 mM Mg-acetate, 100 mM K-glutamate, 2 mM spermidine, 0.05 mM spermine, 1 mM DTT) and layered over a 10–40% sucrose gradient in GF buffer. Samples without RNAse were treated identical, except RNAseA was omitted. Samples were spun at 21,389 g for 18 h in a SW-40 rotor. Gradients were fractionated using a Biocomp piston gradient fractionator and the absorbance at 254 nm recorded. Sucrose density gradient data were plotted using the matplotlib python library.

## Sample preparation for cryo-EM

In vitro translation reactions were performed in the PURE translation system using the PURExpress delta ribosome kit (NEB, #E3313S) as described above, in the absence of $^{14}$C-Val. Translation reactions were supplemented with 0.8 U/μL RNAsin Plus RNase inhibitor (Promega, N261B). SolA, factor mix, and RNAsin Plus were combined on ice, followed by a preincubation at 37 °C for 2 min, and added directly to polysomes (0.7 μM final concentration) that had been preincubated at 37 °C for 2 min. After 1 min reaction time, 4 μL of the reaction mixture were withdrawn, spotted directly onto freshly glow-discharged (20 s at 15 mA, using an easyGlow Discharge Cleaning system (PELCO)) holey-carbon grids (Cu 300 mesh R2/2, without additional carbon (Quantifoil Micro Tools GmbH)), blotted for 1–2 s, and flash frozen in liquid ethane using a Vitrobot Mark IV plunger (ThermoFisher Scientific) after a wait time of 40 s at 4 °C.

## Cryo-EM data acquisition

Cryo-EM data of non-reactivated *E. coli* polysomes were recorded on a 300 kV Tecnai Polara cryo-EM (Thermo Fisher) equipped with a K2 Summit direct electron detector (Gatan) in super-resolution mode at a calibrated pixel size of 0.625 Å/px (31.000x nominal magnification, 50 frames per 10 s movie, 200 ms exposure per frame, total electron dose of 62 e$^-$/Å$^2$ per movie). In total 4520 movies were acquired covering a defocus range from −1 to −2.5 μm. Movies were aligned and dose-weighted using MotionCor2[58]. Cryo-EM data of reactivated *E. coli* polysomes were collected on a Titan Krios G3i transmission electron microscope (ThermoFisher Scientifc) operated at 300 kV equipped with an extra bright field-emission gun (XFEG), a BioQuantum post-column energy filter (Gatan) and a K3 direct electron detector (Gatan). Images were recorded in low-dose mode as dose-fractionated movies using EPU (ThermoFischer Scientific) with a maximum image shift of 9 μm using aberration free image shift (AFI) and fringe-free imaging (FFI). In total 8,983 movies were acquired each for 1.13 s applying a dose rate of 11.7 e$^-$/px/s resulting in a total dose of 45 e$^-$/Å$^2$ distributed over 45 fractions (with an individual dose of 1 e$^-$/Å$^2$ per fraction). Data were recorded in energy filtered zero-loss (slit width 20 eV) nano-probe mode at a nominal magnification of 81.000x, resulting in a calibrated pixel size of 0.53 Å/px on the specimen level in super-resolution mode with a 100 μM objective aperture and defocus values ranging from −0.5 to −2 μm. In each hole, four movies were acquired. Movies were aligned, dose-weighted and binned to a pixel size of 1.06 Å/px in WARP[59].

## Cryo-EM image processing, 3D reconstruction and particle sorting

Defocus values were estimated using Gctf[60]. Templates for particle picking were generated in SPIDER[61]: A density map of a 70S reference model (PDB: 4V9D[62]) was generated and low pass filtered to 20 Å. Subsequently, 84 back projections of the density were generated and 2D-classified into four classes representing averaged particle orientations. The four averages were used as templates for particle picking in Gautomatch (developed by K. Zhang). Particles were extracted and

normalized in Relion 3.0[63] using a box size of 432 and subsequently Fourier cropped to a pixel size of 3.18 Å/px and a box size of 144 for particle sorting. Particles were sorted using cryoSPARC v3.3.1[23]. For an initial round of particle sorting, 30S, 50S, and 70S 3D templates, as well as a non-ribosomal template were generated internally from the present data using a combination of ab initio classification, heterogenous refinement, and 3D variability-based classification[64]. Internally generated non-ribosomal, 30S, 50S, and 70S 3D templates were used for an initial round of heterogenous refinement. Particle classes were then further sorted using global (a global mask was used) 3D variability-based classification yielding 30S PIC, 50S and 70S particle populations. Pre-sorted 70S particles were locally refined with a 50S mask. Different local masks were for subsequent rounds of focused 3D variability-based classification: a mask encompassing the 30S was used for classification focused on the 30S, a mask encompassing EF-Tu and A/T-tRNA was used for classification focused on the factor binding site, a mask encompassing RF1/2 was used for classification focused on the RF1/2 binding site. Sorted 70S functional states were subjected to an additional round of global 3D variability to assess complex homogeneity. Consistently for all 70S functional states, global 3D variability indicated further heterogeneity in the region around bS1. A local mask encompassing bS1 and the region around the mRNA exit was used for a final round of focused 3D variability-based classification. Particles were split into classes containing either density for bS1 or a neighboring 30S. Particles exhibiting the additional 30S density were re-extracted at a larger box size (pixel size: 3.18 Å/px, box size 234). Each of the re-extracted disome particles (pre-sorted based on leading 70S functional state) was subjected to a local refinement focused on the queueing 70S followed by 3D variability-based classification focused on the queueing A-site region. Classification yielded three distinct functional states for queueing ribosomes: closed non-rotated PRE, rotated PRE, and POST states. Prior to high resolution refinements of leading 70S, queueing 70S, and the disome interface, all closed non-rotated PRE leading 70S particles, all closed non-rotated PRE queueing 70S particles, and all non-rotated disome particles were refined locally (using leading, queueing, and interface masks, respectively) and subjected to another round of focused 3D variability-based classification. The final particle populations were refined using a pixel size of 1.06 Å/px. Non-rotated and rotated trisome complexes were isolated from total non-rotated and rotated disome populations using 3D variability-based classification focused on the region downstream of the leading 70S.

### Atomic model building and refinement

Atomic models of previously reported 70S functional states (PDB IDs: 5WFK[19], 6WDJ[20], 6WDL[20], 7N30[21], 7N2U[21], 7N31[21], and 6OT3[22]) were rigid body docked into corresponding $70S_L$ and $70S_Q$ local maps using Coot 0.9.6[65]. Subsequently, model-to-map cross-correlations and cross-resolutions were calculated in PHENIX[66] using the phenix.validation_cryoem tool. For model building of the elongating disome complex, the atomic model of the *E. coli* 70S ribosome (PDB ID: 7N1P[21] was initially docked into the postprocessed density maps with UCSF Chimera[67] and manually adjusted in Coot 0.9.6[65]. As the modeled mRNA represents a mixture of all isolated native *E. coli* mRNAs bound to polysomes, a random sense mRNA sequence was chosen, except for the three codons, which were adjusted to the anticodons. The tRNAs were selected based on the prevalence of tRNAs found in *E. coli* disomes upon pausing[35]. For $70S_Q$ these were: $tRNA_{Ser}$ (serV; anticodon: GCU, codon: AGC) for E-site, $tRNA_{Phe}$ (pheU; anticodon: UUU, codon: GAA) for P-site and $tRNA_{Val}$ (valT; anticodon: GCU, codon: AGC) for A-site. For $70S_L$ these were: $tRNA_{Arg}$ (argV; anticodon: ICG, codon: CGC) for E-site, $tRNA_{Trp}$ (trpT; anticodon: CCA, codon: UGG) for P-site and $tRNA_{Ala}$ (alaW; anticodon: GGC, codon: GCC for A-site. A (poly-Ala)-Thr-Ala-Ser-Phe-Val polypeptide was modeled as nascent chain into the queueing ribosome. For the pre-attack state of the leading

ribosome, a (poly-Ala)-Val-Arg-Asn-Ala-Arg-Trp polypeptide was modeled, which was linked to the P-site $tRNA_{Trp}$. Nucleotide A76 of the aminoacylated A-site $tRNA_{Ala}$ was linked to alanine. The model for protein bS1 was generated using AlphaFold2[24] and manually placed and adjusted in Coot. Models of the disome interface classes were built by rigid body docking of the queuing and leading ribosome models in Coot 0.9.6[65] into pre-aligned interface maps. For this purpose, refined interface maps were aligned with the 16S $rRNA_Q$ of the non-rotated interface class 1 in ChimeraX[67]. First, the 16S $rRNA_Q$ of the non-rotated interface model (PDB: 8R3V) was rigid body docked into the non-rotated interface class 1 map using Coot 0.9.6[65]. Second, using the 'molmap' command in ChimeraX[67] a 3 Å map of the 16S $rRNA_Q$ was generated. Subsequently, non-rotated class 2, as well as rotated interface classes 1 and 2 maps were then aligned with the generated 16S $rRNA_Q$ map using the 'fit in map' command in ChimeraX[67] and resampled using the 'volume resample' command. During building of the rotated interface models, hybrid tRNA positions were adjusted with the guidance of a pre-translocation 70S ribosome structure (PDB: 7SSN[68]). All deposited models were refined over multiple rounds using the module 'phenix.real_space_refine' in PHENIX[66] and interactive model building and refinement in Coot, using libG restraints[69] for the RNAs. Ligand restraints were generated using phenix.eLBOW. The quality of all refined models, summarized in Supplementary Table 1, was assessed using the 'comprehensive model validation' function in PHENIX, the wwPDB validation server[70] and MolProbity[71].

### Structure analysis and presentation

**Presentation of cryo-EM densities and atomic models.** All visualizations of cryo-EM densities and atomic models were generated using UCSF ChimeraX[67].

**$bL9$ CTD blocks elongation factor binding to $70S_Q$.** Overlay of $bL9_L$ bound to $30S_Q$ and EF-G·GTP was generated by fitting the structure of an EFG·GTP bound 70S translocation intermediate (PDB: 7N2V) into the map of $70S_Q$. Subsequently, a 10 Å map was generated from the fitted EF-G·GTP model and shown as transparent density.

**$bS1_Q$ N-terminal alpha helix overlaps with proposed SmrB binding site on $30S_Q$.** Superimposition of SmrB on $70S_Q$ was done by aligning the model of the SmrB bound disome rescue complex (PDB: 7QGR)[6] with the present $30S_Q$ structure.

**$bL9_L$ CTD and 16S $rRNA_Q$ binding plasticity.** For comparison of the *E. coli* crystal structures 4V4Q[49], 4YBB[48] and 6BY1[50] with the present disome structure, PDB models were aligned with the 16S rRNA of the $30S_Q$ of the present disome structure. The distance between Cα of Phe91 in 6BY1 (int1) and Cα of Phe91 in 4V4Q were measured using the distance tool in UCSF ChimeraX[67].

**Figure preparation.** Data were plotted using Microsoft Excel. Figures were assembled using Adobe Illustrator.

### Reporting summary

Further information on research design is available in the Nature Portfolio Reporting Summary linked to this article.

## Data availability

The atomic coordinates and electron density maps generated in this study have been deposited in the Protein Data Bank (https://www.rcsb.org/) and in the EM DataResource (https://www.emdataresource.org/) with the accession numbers 8PKL / EMD-17743 (leading ribosome), 8PEG / EMD-17631 (queueing ribosome), 8R3V / EMD-18875 (non-rotated disome focused on the interface), 8RCL / EMD-19054 (non-rotated disome interface class 1), 8RCM / EMD-19055 (non-rotated disome interface class 2), 8RCS / EMD-19058 (rotated disome interface

class 1), 8RCT / EMD-19059 (rotated disome interface class 2). The electron density maps of the 24 distinct disome states as well as the non-rotated and rotated trisome states were also deposited in the EM DataResource. Their accession numbers are stated in Supplementary Table 1. Published structural data used and referenced in this article were obtained from Protein Data Bank under codes 4V9D, 4YBB, 4V4Q, 6BY1, 6WDD, 7N1P, 7N2V, 7N2C, 7N30, 7SSN, 7QGR, 8CVJ and from AlphaFold2 Protein Structure Database under code P0AG67. Source data are provided with this paper.

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

## Acknowledgements

We thank Simon Lauer, Franziska Wiechert, Justus Loerke, and Martin Heck for intensive scientific discussions and excellent suggestions and Helena Seibel for technical assistance. Y.V.G.S. acknowledges funding support from the Deutsche Forschungsgemeinschaft (DFG) through DFG grant GU 1891/1-1 (to Y.V.G.S.). We thank the Core Facility for cryo-Electron Microscopy (CFcryoEM) of the Charité—Universitätsmedizin Berlin for support in acquisition of the data. The CFcryoEM was supported by the German Research Foundation (DFG) through grant No. INST 335/588-1 FUGG. T.F. acknowledges funding support from the Fonds der Chemischen Industrie through the Kekulé fellowship.

## Author contributions

A.U., M.D. and B.S. prepared polysome samples. Y.V.G.S. performed translation and nuclease protection assays. Y.V.G.S. and B.S. performed sucrose gradients. Y.V.G.S. and T.F. prepared samples for cryo-EM; J.B., T.M., C.A.D. and T.S. collected cryo-EM data; T.F. processed, analyzed, and visualized structural data. M.S. built and refined atomic models; T.F., M.S., Y.V.G.S. and C.M.T.S. interpreted the structural data. Y.V.G.S. and C.M.T.S. designed and supervised the study. T.F. wrote the manuscript. M.S., Y.V.G.S. and C.M.T.S. edited the manuscript. All authors discussed the results and provided input for the manuscript.

## Funding

## Competing interests

The authors declare no competing interests.
