## [Peer Review File · Nature Communications]

Transient disome complex formation in native polysomes during ongoing protein synthesis captured by cryo-EMREVIEWER COMMENTS

Reviewer #1 (Remarks to the Author):

Ribosomes collisions trigger rescue mechanisms in bacteria via specific protein factors. Two of the established mechanisms involve tmRNA/SmrB system and MutS2 ATPase. Cryo-EM studies of native complexes of stalled and translating ribosome collisions provided mechanistic insights into those pathways: 10.1038/s41586-022-04416-7, 10.1038/s41586-022-04487-6. The reported structural features in those studies highlighted the importance of ribosome collisions for rescue. Another study (10.1038/s41586-022-05255-2) focused on translating ribosomes in cells using cryo-ET and found that association into polysomes is mediated by the ribosomal protein L9, which extended conformation mitigates collisions to facilitate translation fidelity.

In the present study, translating ribosomes collisions have been characterised by single-particle cryo-EM. The experimental design differs from the previous studies as it aims translating ribosomes associated with each other and not rescue mechanisms. The work doesn't investigate ribosomal complexes in cells, but rather polysomes have been purified from bacteria, followed by addition of components of in-vitro translation system to stimulate activity. This direction has a potential, and it hasn't been exhaustively explored in previous works due to the limited resolution of cryo-ET. The manuscript deals with only the structure, and the biological implication is speculated based on the produced structural data. This approach is valid as long as the quality of structural data permits derivation of a biological insight. Therefore, the current review is focused on technical aspects of the structural studies in order to help the authors to bring the study to a publishable level.

Limitations of the study:

- The main finding of this work as described in the abstract is that the ribosomal protein L9 adopts a stretched conformation in polysomes to block factor-binding in the following ribosome that prevents aberrant protein synthesis. This conclusion is identical to the mentioned previous works.

10.1038/s41586-022-05255-2: 'We therefore propose that within tightly assembled polysomes, L9 of one ribosome can adopt an extended conformation that sterically interferes with elongation factor binding to the following ribosome. This local coordination mechanism can buffer adjacent ribosomes and help to maintain translation fidelity by avoiding direct collision within polysomes during active translation elongation.'

10.1038/s41586-022-04487-6: 'bL9 is specifically recruited to the leading ribosome 50S subunit near the L1 stalk via its N-terminal half, while the C-terminal half bridges over to the collided ribosome and contacts the 30S subunit 16S rRNA, where it would clash with the mRNA-translocating elongation factor G (EF-G), thereby locking the collided ribosome in a rotated state. Notably, bL9 suppresses translational frameshifting caused by ribosome collisions. Thus, in locking the collided ribosome in a rotated state, bL9 is probably important to prevent frameshifting caused by the strained inter-ribosomal mRNA.'

- Figure 3 illustrating the L9 theme is also nearly identical to the published data, see 10.1038/s41586-022-04487-6 Figures 2, and ED6 ('The C-terminal half of bL9 sterically excludes binding of EF-G on the collided ribosome').

- The density shown in Figure 3 doesn't support the modelling of the residues, most of which are not covered by the map.

- The resolved polysomes used for Figure 1 do not seem to contain translation factors, at least no coordinates are not reported for those. This substantially compromises the analysis presented in Figure 1, which implies activity of translation factors.

- The analysis is performed on reconstituted samples, but the resolution for most of the classes shown in Figure 2 is limited (4.9 – 9.3 Å). No models have been built for any of those 24 cryo-EM reconstructions. Therefore, it's impossible for a reader to assess the models and put them in the context of translation.

- The data shown in Figures 5 and 6 relies on authors' interpretation of low resolution reconstructions, and they make conclusions regarding the function of the L1 stalk and bL9. Again, no models are provided.

Given those issues, the following points might be helpful to bring this work towards a publication:

- Resolve the disome to a resolution level that can provide a new insight and in presence of translation factors, at least as in the previous report.

- Importantly, all the data should be supported with models. Please improve and resolve the 24 low resolution reconstructions, so that they can be modelled to derive biological conclusions.

- At the moment, most of the interpretations are speculative. Using delta L9 strain would help to provide a reference for the structural analysis, and describe the mechanism more in depth.

- Rewrite the manuscript, so that it focuses on new findings, and not repetitive with previous reports. Credit should be given to previous studies when the same findings are reported.

- Please avoid terminology such as 'near-atomic', 'multi particle', as well as subjective descriptions such as 'high-resolution'. Instead, indicate for every discussed feature the local resolution in that region, and also write it in the corresponding figure panels.

Reviewer #2 (Remarks to the Author):

In their manuscript 'Transient disome complex formation in native polysomes during ongoing protein synthesis captured by cryo-EM' Flügel et al analyze polysomes from *E.coli* ex vivo using cryo-EM single particle analysis. Polysomes subjected to gel filtration did display different ribosomal configuration, yet association with factors such as EF-G and EF-Tu was barely observed, which differs from in situ observations. To re-populate elongation-factor bound intermediates the authors re-constituted the isolated polysomes with the PURE system, which resulted in continuation of translation for ~5 minutes. Indeed, the authors succeeded in capturing EF-G and RF1/2 containing intermediates under these conditions. While this strategy does not capture fully native ribosome intermediates, the appeal of the approach is that high-resolution snapshots can be obtained, and it offers the possibility to study re-activated translation under chemically defined conditions. Focused classification and refinement provide detailed insights into the interface of leading and trailing ribosome in disomes, with the protein bL9 emerging as a key factor in this interaction.

The results of this study are complementary to previous cryo-ET results obtained in situ for translating polysomes in *M. pneumoniae* (Xue Nature 2022). Both studies suggest the same role for (b)L9, which is to sterically hinder recruitment of factors by following / queuing ribosome to avoid collisions. Compared to the cryo-ET study, Flügel et al provide a deeper classification and detailed interface analysis. The authors describe 4 additional inter-ribosome contacts that form in disomes. The study also complements recent cryo-EM SPA studies of in vitro stalled disomes (Saito et al Nature 2022 and Cerullo et al Nature 2022). These stalled disomes show a similar architecture allowing to dissect the detailed differences in bL9 in colliding disomes compared to compact – likely pausing - disomes.

Overall, the analysis of the structural data is very thorough, the manuscript is very well written and its visualizations convey the molecular concepts very well. The main shortcoming may be that the novelty of the conclusions are less clear. For example, the abstract reads: "Our data show that bL9 adopts a stretched conformation upon disome formation and blocks the factor-binding site of the queueing ribosome, suggesting that bL9 stalls queueing ribosomes to thwart harmful collisions." For comparison, in Xue et al (Nature 2022) the results and discussion conclude with: "We therefore propose that within tightly assembled polysomes, L9 of one ribosome can adopt an extended conformation that sterically interferes with elongation factor binding to the following ribosome. This local coordination mechanism can buffer adjacent ribosomes and help to maintain translation fidelity by avoiding direct collision within polysomes during active translation elongation." Thus, the recommendation of this reviewer is to emphasize the significance of the novel findings, such as the 5 inter-ribosome contacts and the reconstitution of disome intermediates in the PURE system.

Specific points:

- Fig S5 appears to be mentioned in the text before S4.

- L. 209 and Fig S4A: ‘Remarkably, the filtered density readily accommodates the three N-terminal bS1 domains.’ While the suggested model is plausible, the fit of the alphaFold model does not appear to add much support, as the density is simply of insufficient resolution for a meaningful correlation, presumably due to flexibility.
- in Fig S4 d and e, on the figure and in legend there seem to be typo with rotated vs. non-rotated.
- L. 289: the ref should likely point to Fig S5a.

Reviewer #3 (Remarks to the Author):

Flügel et al. used ex-vivo-derived bacterial polysomes for in vitro translation and then analyzed the ribosome structures by cryo-EM. This study revealed a wide variety of disome and trisome structures in bacteria, highlighting the pivotal role of bL9 to stabilize ribosome collisions. This reviewer recommended addressing the following points before publication.

Major points:

1. The manuscript could be further strengthened by the examination of the potential roles of bL9 in ribosome collisions in bacteria. Authors could address this by partial RNase treatment and the subsequent sucrose density gradient (as conducted in Guydosh et al. Cell 2014) with point mutations or deletion of bL9.

Minor Points:

1. Table S1 is incomplete: it missed a lot of stats that should be filled.
2. The legend of Figure S2 does not contain “b” but instead has “c” and “d”.
3. Authors found that leading ribosomes were stalled due to slow peptidyl transfer reaction. These data were consistent with the earlier biochemical studies that showed that many bacterial ribosome stall sites were resistant to puromycin treatment (Chadani et al. PNAS 2016; Fujita et al. RNA 2022). The authors should consider citing these works in the discussion.
4. Regarding the citations for disome profiling at line 301, please consider citing these works too: PMID: 32375038, PMID: 32703885, and PMID: 32615089.

Point-by-point response to the reviewers' comments

Reviewer #1:

Ribosomes collisions trigger rescue mechanisms in bacteria via specific protein factors. Two of the established mechanisms involve tmRNA/SmrB system and MutS2 ATPase. Cryo-EM studies of native complexes of stalled and translating ribosome collisions provided mechanistic insights into those pathways: 10.1038/s41586-022-04416-7, 10.1038/s41586-022-04487-6. The reported structural features in those studies highlighted the importance of ribosome collisions for rescue. Another study (10.1038/s41586-022-05255-2) focused on translating ribosomes in cells using cryo-ET and found that association into polysomes is mediated by the ribosomal protein L9, which extended conformation mitigates collisions to facilitate translation fidelity.

In the present study, translating ribosomes collisions have been characterised by single-particle cryo-EM. The experimental design differs from the previous studies as it aims translating ribosomes associated with each other and not rescue mechanisms. The work doesn't investigate ribosomal complexes in cells, but rather polysomes have been purified from bacteria, followed by addition of components of in-vitro translation system to stimulate activity. This direction has a potential, and it hasn't been exhaustively explored in previous works due to the limited resolution of cryo-ET. The manuscript deals with only the structure, and the biological implication is speculated based on the produced structural data. This approach is valid as long as the quality of structural data permits derivation of a biological insight. Therefore, the current review is focused on technical aspects of the structural studies in order to help the authors to bring the study to a publishable level.

Limitations if the study:

The main finding of this work as described in the abstract is that the ribosomal protein L9 adopts a stretched conformation in diosomes to block factor-binding in the following ribosome that prevents aberrant protein synthesis. This conclusion is identical to the mentioned previous works. 10.1038/s41586-022-05255-2: 'We therefore propose that within tightly assembled polysomes, L9 of one ribosome can adopt an extended conformation that sterically interferes with elongation factor binding to the following ribosome. This local coordination mechanism can buffer adjacent ribosomes and help to maintain translation fidelity by avoiding direct collision within polysomes during active translation elongation.'

We thank reviewer #1 for their remarks and for appreciating the perceived potential of our work. However, we disagree with the reviewer's comment saying that our main findings are identical to previous works. We admit that that our initial manuscript (especially the abstract) was too narrowly focused on bL9 and that we did not effectively communicate our study's insights. In the following, we attempt to better convey our findings. We thoroughly revised our manuscript and shifted its focus from bL9's interaction with collided ribosomes to a deeper analysis of disome interface dynamics during elongation.

The reviewer rightfully points out that the cryo-ET study by Xue et al. illustrated how bL9's CTD obstructs the factor binding site of queueing ribosomes to thwart harmful collisions. However, the previous cryo-ET maps were limited in resolution, which restricted the analysis to the domain level. Our findings go beyond the structural insights presented by Xue et al. as we visualize the key residues involved in the bL9 interaction at the sidechain level. Importantly, we present 24 structures of

functionally distinct disomes complexes (Fig. 2), some of which are bound by translation factors. Our analysis for example includes an unperturbed pre-nucleophilic attack state and an RF1/2-bound termination state in the context of *E. coli* polysomes. Xue et al. neither identified functional states of the presented disome complexes nor did they resolve translation factors bound to disomes. Furthermore, we investigate how elongation dynamics reshape the disome interface and present conformationally distinct disome interface structures (Fig. 5 and S3). These structures show how the bL9 CTD is repositioned upon inter-subunit rotation of the leading ribosome thereby providing new structural information on bL9 binding modes and plasticity.

10.1038/s41586-022-04487-6: 'bL9 is specifically recruited to the leading ribosome 50S subunit near the L1 stalk via its N-terminal half, while the C-terminal half bridges over to the collided ribosome and contacts the 30S subunit 16S rRNA, where it would clash with the mRNA-translocating elongation factor G (EF-G), thereby locking the collided ribosome in a rotated state. Notably, bL9 suppresses translational frameshifting caused by ribosome collisions. Thus, in locking the collided ribosome in a rotated state, bL9 is probably important to prevent frameshifting caused by the strained inter-ribosomal mRNA.'

Here reviewer #1 refers to a study of stalled disomes by the Pfeffer and Joazeiro labs. The conceptual difference is that in our study the leading ribosome was not stalled by an antibiotic to induce disome formation. In contrast to these previous works, we observe transient disome formation under unperturbed conditions. Furthermore, unlike the mentioned study of chloramphenicol-stalled disomes that described that bL9 locks the collided ribosome in a rotated state, our data show that under unperturbed conditions, queuing ribosomes adopt three distinct functional states: classical PRE, rotated-PRE-2, and POST (see Figure 2). As mentioned above, we also show how the bL9 CTD is repositioned upon inter-subunit rotation of the leading ribosome.

- Figure 3 illustrating the L9 theme is also nearly identical to the published data, see 10.1038/s41586-022-04487-6 Figures 2, and ED6 ('The C-terminal half of bL9 sterically excludes binding of EF-G on the collided ribosome').

We do not agree with the reviewer's comment saying that the illustrated bL9 theme is nearly identical to the study by Cerullo et al.. The structure presented by Cerullo et al. was generated almost entirely by rigid-body docking and shows the bL9 theme merely at the domain level. Due to map segmentation and low local resolution, sidechain details of the interaction between bL9's CTD and the 16S rRNA of the collided ribosome are not visualized. Moreover, the presented composite map has been generated using the "Combine Focused Maps" tool in phenix (see method section in 10.1038/s41586-022-04487-6). Thus, inter-ribosomal contacts were not considered due to masking during the local refinement. Indeed, our analysis of the submitted PDB models by Cerullo et al. shows that their approach resulted in clashes between the models of leading and collided ribosomes (see Figure below). Consequently, no meaningful analysis of inter-ribosomal contacts at molecular detail can be achieved.

Analysis of PDB models 7QV1 (leading 70S, shown in blue) and 7QV2 (collided 70S, shown in orange) by Cerullo et al. showing coordinate clashes.

In contrast, through focused refinement of the disome interface as one volume rather than as two individually masked 70S particles, we were able to generate an authentic interface map at near-atomic resolution allowing to visualize sidechain details of the bL9 CTD interaction with h5/15. In Figure 3c we show how F91 of the bL9 CTD stacks with U368 of the 16S rRNA of the queueing ribosome.

- The density shown in Figure 3 doesn't support the modelling of the residues, most of which are not covered by the map.

We agree that some close-ups of the interfaces in Figure 3 were not adequately illustrated. We were able to slightly improve the sidechain details of the interface map by adjusting the applied b-factor for map sharpening. However, showing all interacting residues in a 2D-projection at a single threshold is difficult. For clarity, we removed some sidechains from the close-ups. But the structural information is present in the cryo-EM map.

- The resolved polysomes used for Figure 1 do not seem to contain translation factors, at least no coordinates are not reported for those. This substantially compromises the analysis presented in Figure 1, which implies activity of translation factors.

We are surprised by the reviewer's comment saying that the polysomes in Figure 1 do not seem to contain translation factors. The translation factors (EF-Tu and RF1/2) are colored in red in Figure 1b and the color-code is explained in the Figure description. As for the coordinates, the individual 70S functional states shown in Figure 1 have not been modelled as they resemble known and previously published structures of translation intermediates. Instead, we fitted the corresponding available PDB models into the present disome maps and calculated their cross resolution (see Figure S5). As in-depth modelling of these known structures would not have yielded more structural insights, we focused on the only unknown state, the pre-attack state. Moreover, the focus of the submitted manuscript lies on the structural details of the disome arrangement rather than the analysis of known 70S translation intermediates.

- The analysis is performed on reconstituted samples, but the resolution for most of the classes shown in Figure 2 is limited (4.9 – 9.3 Å). No models have been built for any of those 24 cryo-EM

reconstructions. Therefore, it's impossible for a reader to assess the models and put them in the context of translation.

The maps and the corresponding resolutions shown in Figure 2 were yielded by global map refinements (refined with a global mask encompassing both leading and queueing 70S). The global disome maps were only refined up to a resolution at which leading and queueing 70S are similarly well defined. We observed that refinement algorithms such as Cryosparc's non-uniform refinement tend to focus on either the leading or queueing particle to further push the resolution. However, this results in blurring of the off-focus particle. Given the intrinsic dynamics of the disome complex, global disome refinements have their resolution limitations. We believe that a global map of intermediate resolution that visualizes both particles at similar level of detail is more meaningful than a map with a blurred particle. To improve our analysis and to achieve higher resolution of the disome functional states, we subjected each of the 24 complexes shown in Figure 2 to local refinements. These refinements yielded resolutions ranging from 3.09-4.6 Å for the leading 70S and 3.21-3.3 Å for the queueing 70S (see Figure S5). We have manually built models for the leading 70S pre-attack state (resolved at 3.09 Å), the queueing classical PRE state (resolved at 3.3 Å), and the non-rotated disome interface (resolved at 3.28 Å). As mentioned above, models for the known 70S functional states were fitted into the locally refined maps and their cross resolution was calculated. Additionally, we have built models for non-rotated and rotated interface classes (see Fig.5). In summary, for all 24 disome states we have three overlapping locally refined cryo-EM maps (leading 70S, disome interface, and queueing 70S) that together represent the respective states with much higher resolution than for the globally refined disome maps.

- The data shown in Figures 5 and 6 relies on authors' interpretation of low resolution reconstructions, and they make conclusions regarding the function of the L1 stalk and bL9. Again, no models are provided.

In Figure 5a we show global maps of disome complexes with known functional states for leading and queueing 70S. Inspection of disome complexes with known functional state identities allows us to keep the queueing 70S functional state constant (non-rotated PRE in Figure 5a) while analyzing the transition from non-rotated PRE to rotated PRE state in the leading 70S. We have now further investigated disome interface dynamics and were able to reconstruct two rotated disome interface classes at 4.46 Å and 5.32 Å. These reconstructions provide further insights into the structural rearrangements occurring upon intersubunit rotation of the leading ribosome. We have illustrated these rearrangements in Figure 5b-f (shown models will be uploaded to the PDB).

Figure 6 describes a putative mechanism that is based on the analysis presented in Figure 5. The additional structural analysis of the rotated interface supports the assumption that intersubunit rotation of the leading 70S weakens the bL9 CTD interaction.

Given those issues, the following points might be helpful to bring this work towards a publication:

- Resolve the disome to a resolution level that can provide a new insight and in presence of translation factors, at least as in the previous report.

We would like to point out that we provide several disome structures in the presence of translation factors (see Fig. 2). Through exhaustive local refinements of the disome complex containing a leading pre-attack state and a queueing classical PRE state, we resolved leading and queueing 70S at nominal

resolutions of 3.09 Å and 3.3 Å, respectively. The global map for this complex is resolved at 4.95 Å. We are not aware of any deposited structures resolving individual leading and queueing 70S (in a disome arrangement) or an entire disome (refined globally) at this level of detail or better. For comparison, leading and collided ribosome maps in the paper by Cerullo et al. were resolved at 3.38 Å (EMD-14162) and 3.43 Å (EMD-14164), respectively. In the paper by Saito et al. leading and collided ribosome maps were resolved at 3.97 Å (EMD-13952) and 4.48 Å (EMD-13955), respectively. In addition to leading and queueing ribosome local maps, we present a focused map of the non-rotated disome interface at 3.28 Å resolution. Unlike previous reports, the resolution of our maps allowed us to manually build models into our experimental data. For some of the interfaces, we were able to resolve inter-ribosomal contacts at the sidechain level, most importantly the interaction of F91 of the bL9 CTD with the bases of h5/15 of the queueing 16S rRNA. We have now expanded our structural analysis and present rotated disome interface maps. These maps provide further insights into the bL9 mechanism illustrating that upon inter-subunit rotation of the leading ribosome bL9's CTD moves 8.7 Å up h5/15 of the queueing 16S rRNA (see Figure 5f).

- Importantly, all the data should be supported with models. Please improve and resolve the 24 low resolution reconstructions, so that they can be modelled to derive biological conclusions.

As described above, we have improved the resolutions of the leading and queueing disome maps and interpreted them by docking the corresponding PDB models and calculating their cross resolution (see Figure S5).

- At the moment, most of the interpretations are speculative. Using delta L9 strain would help to provide a reference for the structural analysis, and describe the mechanism more in depth.

The structural analysis of a delta bL9 strain is an interesting suggestion. However, such an analysis is beyond the scope of the present manuscript as its focus lies on the analysis of disome interface dynamics during unperturbed elongation. Nonetheless, as suggested also by reviewer #3 we have now performed nuclease digest experiments using a delta bL9 strain (lines 184-196). The results show that disome and trisome complexes are less RNase-resistant than the wild-type suggesting that bL9 enhances disome complex stability. (see Figure S4)

- Rewrite the manuscript, so that it focuses on new findings, and not repetitive with previous reports. Credit should be given to previous studies when the same findings are reported.

We have rewritten our manuscript and shifted the focus from the mechanism by which bL9 obstructs the factor binding site of the queueing ribosome to the investigation of disome interface dynamics during elongation. The mentioned previous reports are referenced and discussed where appropriate.

- Please avoid terminology such as 'near-atomic', 'multi particle', as well as subjective descriptions such as 'high-resolution'. Instead, indicate for every discussed feature the local resolution in that region, and also write it in the corresponding figure panels.

In our revised manuscript we avoid subjective descriptions of resolution. As suggested, we indicate local resolution where sidechain details are presented. However, “near-atomic resolution” is a generally accepted and widely used term in structural biology and refers to the resolution range between true atomic resolution and the resolution that allows building models de novo (<4 Å). Also, the term “multi particle” has been used in the cryo-EM field for nearly two decades and we do not see a reason why it should be avoided.

Reviewer #2 (Remarks to the Author):

In their manuscript ‘Transient disome complex formation in native polysomes during ongoing protein synthesis captured by cryo-EM’ Flügel et al analyze polysomes from E.coli ex vivo using cryo-EM single particle analysis. Polysomes subjected to gel filtration did display different ribosomal configuration, yet association with factors such as EF-G and EF-Tu was barely observed, which differs from in situ observations. To re-populate elongation-factor bound intermediates the authors re-constituted the isolated polysomes with the PURE system, which resulted in continuation of translation for ~5 minutes. Indeed, the authors succeeded in capturing EF-G and RF1/2 containing intermediates under these conditions. While this strategy does not capture fully native ribosome intermediates, the appeal of the approach is that high-resolution snapshots can be obtained, and it offers the possibility to study re-activated translation under chemically defined conditions. Focused classification and refinement provide detailed insights into the interface of leading and trailing ribosome in disomes, with the protein bL9 emerging as a key factor in this interaction.

The results of this study are complementary to previous cryo-ET results obtained in situ for translating polysomes in M. pneumoniae (Xue Nature 2022). Both studies suggest the same role for (b)L9, which is to sterically hinder recruitment of factors by following / queuing ribosome to avoid collisions. Compared to the cryo-ET study, Flügel et al provide a deeper classification and detailed interface analysis. The authors describe 4 additional inter-ribosome contacts that form in disomes. The study also complements recent cryo-EM SPA studies of in vitro stalled disomes (Saito et al Nature 2022 and Cerullo et al Nature 2022). These stalled disomes show a similar architecture allowing to dissect the detailed differences in bL9 in colliding disomes compared to compact – likely pausing - disomes.

Overall, the analysis of the structural data is very thorough, the manuscript is very well written and its visualizations convey the molecular concepts very well. The main shortcoming may be that the novelty of the conclusions are less clear. For example, the abstract reads: “Our data show that bL9 adopts a stretched conformation upon disome formation and blocks the factor-binding site of the queueing ribosome, suggesting that bL9 stalls queueing ribosomes to thwart harmful collisions.” For comparison, in Xue et al (Nature 2022) the results and discussion conclude with: “We therefore propose that within tightly assembled polysomes, L9 of one ribosome can adopt an extended conformation that sterically interferes with elongation factor binding to the following ribosome. This local coordination mechanism can buffer adjacent ribosomes and help to maintain translation fidelity by avoiding direct collision within polysomes during active translation elongation.” Thus, the recommendation of this reviewer is to emphasize the significance of the novel findings, such as the 5 inter-ribosome contacts and the reconstitution of disome intermediates in the PURE system.

We thank reviewer #2 for the helpful input. We have rewritten our manuscript and shifted the focus from bL9 to a more in-depth investigation of disome interface dynamics and highlighted our novel findings. We highly appreciate the very thorough comments on our manuscript.

Specific points:

- *Fig S5 appears to be mentioned in the text before S4.*

All figures are now arranged in the order in which appear in the text.

- *L. 209 and Fig S4A: 'Remarkably, the filtered density readily accommodates the three N-terminal bS1 domains.' While the suggested model is plausible, the fit of the alphaFold model does not appear to add much support, as the density is simply of insufficient resolution for a meaningful correlation, presumably due to flexibility.*

We now only show the first two domains of bS1 fitted into the filtered queueing 70S density.

- *in Fig S4 d and e, on the figure and in legend there seem to be typo with rotated vs. non-rotated.*

We have corrected the typo in Fig. S4 d and e.

- *L. 289: the ref should likely point to Fig S5a.*

We corrected this Figure reference.

Reviewer #3 (Remarks to the Author):

Flügel et al. used ex-vivo-derived bacterial polysomes for in vitro translation and then analyzed the ribosome structures by cryo-EM. This study revealed a wide variety of disome and trisome structures in bacteria, highlighting the pivotal role of bL9 to stabilize ribosome collisions. This reviewer recommended addressing the following points before publication.

Major points:

- 1. The manuscript could be further strengthened by the examination of the potential roles of bL9 in ribosome collisions in bacteria. Authors could address this by partial RNase treatment and the subsequent sucrose density gradient (as conducted in Guydosh et al. Cell 2014) with point mutations or deletion of bL9.*

We thank reviewer #3 for the excellent experimental suggestion to strengthen our manuscript. As suggested, we have performed RNase digest experiments before and after re-stimulation using a delta bL9 strain and a wild-type strain. The results show that disome and trisome complexes are less RNase-resistant than the wild-type suggesting that bL9 enhances disome complex stability (see Figure S4).

Minor Points:

- 1. Table S1 is incomplete: it missed a lot of stats that should be filled.*

We have completed Table S1.

2. The legend of Figure S2 does not contain “b” but instead has “c” and “d”.

We have corrected the legend of Figure S2.

3. Authors found that leading ribosomes were stalled due to slow peptidyl transfer reaction. These data were consistent with the earlier biochemical studies that showed that many bacterial ribosome stall sites were resistant to puromycin treatment (Chadani et al. PNAS 2016; Fujita et al. RNA 2022). The authors should consider citing these works in the discussion.

We have now cited these works in the discussion (lines 394-396).

4. Regarding the citations for disome profiling at line 301, please consider citing these works too: PMID: 32375038, PMID: 32703885, and PMID: 32615089.

We reference these publications in our revised manuscript (line 371).

REVIEWERS' COMMENTS

Reviewer #2 (Remarks to the Author):

The manuscript has improved considerably upon revision and the RNase digestion experiment strengthened the conclusions.

After re-reading the manuscript this reviewer recommends toning down the statement that the 'non-rotated disome interface structure illustrates at the

sidechain level how residues of the bL9L CTD specifically interact with the 16S rRNAQ' (l. 333-5). While Fig. 3c suggests that large sidechains such as Phe91 are somewhat resolved, the visualization of Lys89 is less compelling, consistent with the lower local resolution limit of the map (3.6 Å). Likewise, the corresponding results section (l. 237) should be reformulated – or the map should be visualized such that it provides better support for the modeled side chain.

After addressing this minor point, this reviewer supports publication.

typo:

- l. 181: 'comprising of': change to 'consisting of' or 'comprising'

Reviewer #3 (Remarks to the Author):

The authors properly addressed the concerns raised in the first round of review. This reviewer recommends this manuscript for publication.

Point-by-point response to the reviewers' comments

Reviewer #2 (Remarks to the Author):

The manuscript has improved considerably upon revision and the RNase digestion experiment strengthened the conclusions.

We are pleased to hear that reviewer #2 finds our revised manuscript considerably improved.

After re-reading the manuscript this reviewer recommends toning down the statement that the 'non-rotated disome interface structure illustrates at the sidechain level how residues of the bL9L CTD specifically interact with the 16S rRNAQ' (l. 333-5). While Fig. 3c suggests that large sidechains such as Phe91 are somewhat resolved, the visualization of Lys89 is less compelling, consistent with the lower local resolution limit of the map (3.6 Å). Likewise, the corresponding results section (l. 237) should be reformulated – or the map should be visualized such that it provides better support for the modeled side chain.

After addressing this minor point, this reviewer supports publication.

We have toned down the statement by removing the phrase "at the sidechain level":

typo:

- l. 181: 'comprising of': change to 'consisting of' or 'comprising'

We have changed "comprising of" to "comprising".

Reviewer #3 (Remarks to the Author):

The authors properly addressed the concerns raised in the first round of review. This reviewer recommends this manuscript for publication.

We are happy to hear that we have properly addressed the concerns of reviewer #3.